# The Role of Sparsity for Length Generalization in LLMs

**Noah Golowich** [1]   **Samy Jelassi** [2]   **David Brandfonbrener** [3]   **Sham M. Kakade** [3]   **Eran Malach** [3]

## Abstract

Training large language models to predict beyond their training context lengths has drawn much attention in recent years, yet the principles driving such behavior of *length generalization* remain underexplored. We propose a new theoretical framework to study length generalization for the next-token prediction task, as performed by decoder-only transformers. Conceptually, we show that length generalization occurs as long as each predicted token depends on a small (fixed) number of previous tokens. Our theoretical model justifies certain techniques to modify positional embeddings which have been introduced to improve length generalization, such as *position coupling*. We support our theoretical results with experiments on synthetic tasks and natural language, which confirm that a key factor driving length generalization is a "sparse" dependency structure of each token on the previous ones. Inspired by our theory, we introduce *Predictive Position Coupling*, which trains the transformer to *predict* the position IDs used in a positional coupling approach. Predictive Position Coupling thereby allows us to broaden the array of tasks to which position coupling can successfully be applied to achieve length generalization.

## 1. Introduction

Enabling large language models (LLMs) to generalize to contexts longer than their training context length has emerged as a key problem in recent years. Indeed, many factors limit the context length of sequences which can be used during training, including the increased *computational cost* of training on long sequences (Tay et al., 2022) as well as the fact that longer sequences may be less numerous in the training *dataset*. Nevertheless, many applications require LLMs to be accurate on extremely long context lengths at inference time: for instance,

a popular technique recently has been to use *scratchpads* or *Chain-of-Thought (CoT)* to perform various logic and reasoning tasks, and the length of the scratchpad can become very large, especially when combined with search or reinforcement learning techniques (DeepSeek-AI et al., 2025; Team et al., 2025).

Unfortunately, transformers struggle to *length generalize* on even very simple arithmetic and logic tasks, such as computing parities, integer addition, and variable assignment (Anil et al., 2022; Kazemnejad et al., 2023). Indeed, only recently have transformer models successfuly been trained to length-generalize to contexts many times their training length on integer addition tasks, using a technique known as *position coupling* (closely related to *Abacus embeddings*) (Cho et al., 2024a;b; McLeish et al., 2024). For many other simple problems, comparable length generalization remains a challenge. In light of this uneven progress, a natural question is whether there is a more principled way of understanding length generalization. *In particular, can we formally reason about what structural properties of data aid or inhibit length generalization? Moreover, can this suggest architectural modifications, such as to positional embeddings, which improve length generalization?*

**Contributions.** In this paper we give a positive answer to the above questions: First, we introduce a class of data distributions, namely those with *sparse planted correlations* (Definition 3.2), which, roughly speaking, captures the property observed in many synthetic and natural language tasks that each token to be predicted depends on a small number $k$ of previous tokens, which we call the *sparsity*. We then establish the following:

- As long as the sparsity $k$ does not grow with the sequence length, then a simple class of models generalizing attention heads (namely, *sparse functional attention*; Definition 3.3) has provable length generalization (Theorem 4.3). This result also relies on an additional assumption of *locality* on the hypothesis class.

- While the locality assumption is typically violated in practice, we show that (a theoretical abstraction of) *position coupling* can remove the locality requirement (Proposition 4.4), thus offering a theoretical justification for this recent technique.

- We perform experiments (Section 5) on *synthetic* and *natural language* data to support our theorical conclusions: for instance, our experiments adjust the sparsity of several synthetic tasks, and we observe that length generalization improves monotonically with decreasing sparsity. For natural language data, we provide evidence that length-generalizing transformers indeed make accurate predictions using a small

---

[1]MIT EECS [2]Harvard University [3]Kempner Institute at Harvard University. Correspondence to: Noah Golowich <nzg@mit.edu>, Samy Jelassi<sjelassi@fas.harvard.edu>, David Brandfonbrener<david.brandfonbrener@gmail.com>, Sham M. Kakade<sham@seas.harvard.edu>, Eran Malach <eran.malach@gmail.com>.

*Proceedings of the 42$^{nd}$ International Conference on Machine Learning*, Vancouver, Canada. PMLR 267, 2025. Copyright 2025 by the author(s).

number of past tokens.

- Inspired by our theory, we introduce a modification of positional coupling, *Predictive Position Coupling*, which, unlike positional coupling, works on tasks for which the coupled position IDs are input-dependent. We show (Section 5.2) that Predictive Position Coupling enables significant length generalization on such tasks.

**Related work.** A few prior works have attempted to give theoretical justifications for the length generalization abilities of transformers (Zhou et al., 2023; Huang et al., 2024; Sabbaghi et al., 2024; Ahuja & Mansouri, 2024); we compare these works, as well as many others, to ours in Appendix A. The idea that *sparsity* and *locality* are important for length generalization has appeared in several of these works (Huang et al., 2024; Sabbaghi et al., 2024). However, none has provided the precise theoretical and empirical evidence which we believe are essential for isolating the importance of these notions.

## 2. Preliminaries

Our focus in this paper is on decoder-only transformers trained to predict the next token. We review the architecture of such transformers in more detail in Appendix B.2. In this section, we discuss a few aspects of the *position encodings* of transformers which have previously been proposed to improve length generalization.

Initially, Vaswani et al. (2017) proposed to introduce positional information in a transformer model using *absolute positional embeddings (APE)* (see also Gehring et al. (2017)). The APE assigns to each token a *position ID* $i$ (typically the position of that token in the sequence) and adds an embedding vector $\mathbf{v}_i^{\mathsf{APE}}$ depending on $i$ to the embedding vector for the corresponding token. Recently, it has become more common to use position encodings which encode *relative positions*. In particular, a popular positional encoding technique in many large-scale open-source transformers is the *rotary positional encoding (RoPE)* (Su et al., 2021), which adjusts the computation of attention scores as follows: it multiplies each key embedding vector with position ID $i$ by a rotation matrix depending on $i$, and each query embedding vector with position ID $j$ by a rotation matrix depending on $j$. The effect of these rotations is that the attention score for a (key,query) pair with position IDs $(i,j)$ depends only on $i-j$ (and not $i$ or $j$ individually).

An obstacle to length generalization in transformers is that the joint distribution of particular tokens and their position IDs seen in training sequences may not match that seen in the longer sequences at test-time. To account for this discrepancy, a common empirical technique is to modify the position IDs at training or test time. As discussed in Section 1, the *positional coupling* technique, which has recently paved the way for major improvements in length generalization in arithmetic and logic tasks, plays a key role in our theoretical and empirical results.

We discuss it next; additional techniques to modify position IDs to achieve length generalization are discussed in Appendix A.2.

### 2.1. Position Coupling

The technique of *position coupling* (Cho et al., 2024a;b; McLeish et al., 2024) (similar to *Abacus* in McLeish et al. (2024)) works specifically for problems with structured input where there is a clear one-to-one relationship between certain tokens. In particular, it assigns each token in a sequence a particular position ID in a way so that tokens assigned the same position ID should have a (task-dependent) "one-to-one correspondence". For example, to solve *string reversal*, i.e., predict the last $L$ tokens of the sequence $\mathbf{X}_1, \ldots, \mathbf{X}_L, \texttt{<SEP>}, \mathbf{X}_L, \ldots, \mathbf{X}_1$, then since the $i$th-to-last reversed token is equal to the $i$th input token, we feed the following position IDs: $1,2,\ldots,L,0,L,\ldots,2,1$.

## 3. Theoretical model

In this section, we formally define our theoretical model. Our primary inspiration is the class of decoder-only transformers, which are trained to predict each next token as a function of the preceding ones in a given sequence. Accordingly, our theoretical framework will focus on the *next-token prediction task*, where we fix lengths $L < \bar{L}$, and attempt to show that models *trained* on sequences of length $\leq L$ to predict a *single token* (interpreted as the $(L+1)$th token in the sequence) also succeed at predicting a *single token* (interpreted as the $(\bar{L}+1)$th token) when *tested* on sequences of length $\bar{L}$.

Fix a set $\mathcal{V}$ which denotes the set of tokens. $\mathcal{V}^\star = \bigcup_{\ell \geq 0} \mathcal{V}^\ell$ denotes the set of all arbitrary-length sequences of tokens. For technical reasons discussed below, we model the problem of predicting the *embedding* (i.e., *representation*) of the next token, as opposed to the token itself, given a sequence of previous tokens. Accordingly, we fix a label set $\mathcal{Y}$ which is a convex subset of $\mathbb{R}^d$ for some $d \in \mathbb{N}$ denoting the space of possible token embeddings, and a loss function $\mathcal{L} : \mathcal{Y} \times \mathcal{Y} \to [0,1]$ with $\mathcal{L}(Y,Y) = 0$ for all $Y \in \mathcal{Y}$. For simplicity, we fix an arbitrary norm $\|\cdot\|$ on $\mathbb{R}^d$, and assume that $\mathrm{diam}(\mathcal{Y}) \leq 1$ with respect to this norm and that $\mathcal{L}(Y,Y') = \|Y - Y'\|$. For a set $\Omega$ and $k \in \mathbb{N}$, we let $\mathrm{Sets}_k(\Omega)$ be the set of size-$k$ subsets of $\Omega$.

A *distribution ensemble* $\mathcal{P}$ is a sequence of distributions $\mathcal{P}_1, \mathcal{P}_2, \ldots$, where for each $\ell \in \mathbb{N}$, $\mathcal{P}_\ell \in \Delta(\mathcal{V}^\ell \times \mathcal{Y})$ represents a distribution over labeled input sequences of length $\ell$. Let $\mathcal{H} \subset \mathcal{Y}^{\mathcal{V}^\star}$ denote a class of functions $h : \mathcal{V}^\star \to \mathcal{Y}$. The *length generalization* problem is as follows: we aim to choose a hypothesis $\hat{h} \in \mathcal{H}$ which enjoys small square loss $\mathcal{L}(\cdot)$ for inputs drawn from $\mathcal{P}_{\bar{L}}$ for some $\bar{L} \in \mathbb{N}$, if we are only allowed to choose $\hat{h}$ based off of inputs of lengths less than $\bar{L}$, i.e., those drawn from $\mathcal{P}_\ell$ for $\ell < \bar{L}$.

How can we use inputs drawn from such $\mathcal{P}_\ell$ to choose $\hat{h}$? A classic paradigm in learning theory is *empirical risk*

*minimization* (see e.g., Shalev-Shwartz & Ben-David (2014)), which stipulates to choose a hypothesis in $\mathcal{H}$ to minimize the average loss over *samples* drawn from $\mathcal{P}_\ell$ for some values of $\ell$. However, as our goal is specifically to understand the *out-of-distribution* generalization behavior from inputs of length $\ell < \bar{L}$ to inputs of length $\bar{L}$, we instead assume that $\hat{h}$ is chosen so as to minimize the *population risk* for inputs of length $\ell < \bar{L}$. This choice allows us to avoid having to consider the (in-distribution) generalization error for samples drawn from $\mathcal{P}_\ell$ ($\ell < \bar{L}$). Formally, we define length generalization as follows:

**Definition 3.1** (Length generalization). For $L, \bar{L} \in \mathbb{N}$, $\epsilon \in (0,1)$, we say that the class $\mathcal{H}$ has $(L, \bar{L}, \epsilon)$-*length generalization with respect to the ensemble* $\mathcal{P}$ if the hypothesis

$$\hat{h} := \operatorname*{argmin}_{h \in \mathcal{H}} \mathbb{E}_{\ell \sim \mathrm{Unif}([L/2, L])} \mathbb{E}_{(\mathbf{X}, \mathbf{Y}) \sim \mathcal{P}_\ell} [\mathcal{L}(h(\mathbf{X}), \mathbf{Y})] \quad (1)$$

satisfies $\mathbb{E}_{(\mathbf{X}, \mathbf{Y}) \sim \mathcal{P}_{\bar{L}}} [\mathcal{L}(\hat{h}(\mathbf{X}), \mathbf{Y})] \le \epsilon$.[1]

For a hypothesis class $\mathcal{H} \subset \mathcal{Y}^{\mathcal{V}^\star}$ and $\delta \in (0,1)$, we say that the ensemble $\mathcal{P}$ is $\delta$-*approximately* $\mathcal{H}$-*realizable* if there is some $h^\star \in \mathcal{H}$ so that $\mathbb{E}_{(\mathbf{X}, \mathbf{Y}) \sim \mathcal{P}_\ell} [\mathcal{L}(h^\star(\mathbf{X}), \mathbf{Y})] \le \delta$ for each $\ell \in \mathbb{N}$. We say that $\mathcal{P}$ is $\mathcal{H}$-*realizable* in the case $\delta = 0$.

### 3.1. Distributional assumptions: sparse structure

Without distributional assumptions on $\mathcal{P}_\ell$, achieving length generalization per Definition 3.1 becomes degenerate in the following manner, even if we assume $\mathcal{H}$-realizability. For any fixed (realizable) choice of the distributions $\mathcal{P}_\ell$ for $\ell < \bar{L}$, unless $\hat{h}$ is *identically equal* to $h^\star$, we can choose $\mathcal{P}_{\bar{L}}$ to have all its mass on some sequence $\mathbf{X} \in \mathcal{V}^{\bar{L}}$ for which $\mathcal{L}(h^\star(\mathbf{X}), \hat{h}(\mathbf{X})) \gg 0$. This choice prevents the loss of $\hat{h}$ defined in (1) from being small, thus ruling out length generalization, as formalized in Proposition B.1.

While some works (e.g., Ahuja & Mansouri (2024); Huang et al. (2024)) have offered explanations for length generalization by showing that in fact, $h^\star$ is learned *exactly* under appropriate assumptions, there is ample empirical evidence (Zou et al., 2023; Wei et al., 2023; Andriushchenko et al., 2024) that transformers can err on worst-case inputs. Accordingly we ask: *Are there distributional assumptions which enable us to establish length generalization in settings where the ground truth hypothesis $h^\star$ may* not *be learned (nearly) exactly in the sense that* $\sup_{\mathbf{X} \in \mathcal{V}^\star} \mathcal{L}(h^\star(\mathbf{X}), \hat{h}(\mathbf{X}))$ *is small?*

The sparse attention patterns in many transformers trained on natural language (e.g., Child et al. (2019); Tay et al. (2022)) suggest that for modeling a wide spectrum of natural language, one needs to attend to only a small number of previous tokens to predict each successive token. Inspired by this observation, we define the following class of distribution ensembles, namely those with *sparse planted correlations*. Roughly speaking,

sequences drawn from such distributions have most of their tokens drawn independently from some distribution $\mu \in \Delta(\mathcal{V})$ (which we think of as a "background distribution", representing tokens not relevant for the task at hand), and a small number $k$ of tokens drawn from some "planted" correlated distribution over $k$-tuples of tokens, denoted by $\mathcal{Q}^{\mathsf{voc}}$ in Definition 3.2 below. One should interpret these planted $k$ tokens as having "relevant information" for the task of predicting the label $\mathbf{Y}$. The particular location of these tokens is drawn independently (denoted $\mathcal{Q}_\ell^{\mathsf{pos}}$ below). Formally, we have:

**Definition 3.2** ($k$-sparse planted correlations). Fix a positive integer $k \in \mathbb{N}$. We say that a distribution ensemble $\mathcal{P} = (\mathcal{P}_\ell)_{\ell \in \mathbb{N}}$ has $k$-*sparse planted correlations* if there are distributions $\mu \in \Delta(\mathcal{V})$, $\mathcal{Q}_\ell^{\mathsf{pos}} \in \Delta(\mathrm{Sets}_k([\ell]))$ for $\ell \in \mathbb{N}$, $\mathcal{Q}^{\mathsf{voc}} \in \Delta(\mathcal{V}^k)$, and a function $g^\star : \mathcal{V}^k \to \mathcal{Y}$ so that the following holds. For each $\ell \in \mathbb{N}$, a sample $(\mathbf{X}, \mathbf{Y}) \sim \mathcal{P}_\ell$ may be drawn as follows: first, we draw $S^\star \sim \mathcal{Q}_\ell^{\mathsf{pos}}, \mathbf{Z} \sim \mathcal{Q}^{\mathsf{voc}}$, and we set:

$$\mathbf{X}_{S^\star} = \mathbf{Z}, \qquad \mathbf{X}_i \sim \mu \ \forall i \notin S^\star, \qquad \mathbf{Y} = g^\star(\mathbf{Z}) \quad (2)$$

While the assumptions in Definition 3.2 that (a) the remaining tokens $\mathbf{X}_i$ for $i \notin S^\star$ are i.i.d. from $\mu$ and (b) the tuple $S^\star$ which indexes the correlated tokens that "matter" is drawn from a fixed distribution $\mathcal{Q}_\ell^{\mathsf{pos}}$ are not realistic, we emphasize that they are made to simplify the proofs and ensure a simple model which captures the salient features that enable length generalization. We leave generalizations, such as that the $\mathbf{X}_i$ are drawn from a Markov chain or Hidden Markov Model, for future work.

An example of a distribution ensemble satisfying Definition 3.2 is the distribution of tokens in the $k$-*gram retrieval task*, of outputting the token following a certain $k$-gram in a given sequence $\mathbf{X}$. In particular, the $k$-gram in question together with the following token form the "planted sequence"; see Appendix B.1 for a formal description.

### 3.2. Defining the hypothesis classes

In this section, we define a hypothesis class which allows us to effectively model the salient aspects of transformers while permitting us to obtain provable length generalization. In particular, we aim to satisfy the following two desiderata:

1. First, we would like there to be many ensembles $\mathcal{P}$ with sparse planted correlations which are realizable with respect to $\mathcal{H}$.

2. Second, we would like the class $\mathcal{H}$ to capture simple transformers (i.e., a single attention head).

To motivate how to arrive at our function class starting from the above criteria, we recall the structure of a single attention head (see Appendix B.2 for a more complete presentation). It takes as input *embedding vectors* $\mathbf{h}_1, ..., \mathbf{h}_L \in \mathbb{R}^d$ which could be, for instance, the results of an embedding matrix multiplied by one-hot encodings of tokens $\mathbf{X}_1, ..., \mathbf{X}_L \in \mathcal{V}$. For some matrices $\mathbf{K}, \mathbf{V} \in$

---

[1]The lower bound of $L/2$ in the interval $[L/2, L]$ from which $\ell$ is sampled is unimportant; any constant factor of $L$ will suffice.

$\mathbb{R}^{d \times d}$ and a query vector[2] $\mathbf{q} \in \mathbb{R}^d$, the attention head computes

$$\widetilde{\mathsf{AttHead}}_{\mathbf{K},\mathbf{V},\mathbf{q}}(\mathbf{h}_1,...,\mathbf{h}_L) = \sum_{i=1}^{L} \alpha_i \cdot \mathbf{V}\mathbf{h}_i \qquad (3)$$

where $\alpha = \mathrm{softmax}(\langle \mathbf{q}, \mathbf{K}\mathbf{h}_1 \rangle, ..., \langle \mathbf{q}, \mathbf{K}\mathbf{h}_L \rangle)$. The softmax over linear functions of $\mathbf{h}_1,...,\mathbf{h}_L$ allows $\widetilde{\mathsf{AttHead}}$ to attend to individual positions $i \in [L]$ whose tokens satisfy some property (e.g., the embedding vector points in a particular direction of $\mathbb{R}^d$, which could represent a certain meaning of the token). Suppose we were to take $\mathcal{H}$ to be the class of attention heads and try to satisfy Item 1: we would need attention heads to be able to apply some function ($g^\star$ in the context of Definition 3.2) to a certain *sub-sequence of $k$ tokens* ($\mathbf{X}_{S^\star}$ in Definition 3.2).[3]

Unfortunately, a single attention head as in (3) is not able to do so for large classes of $g^\star$, when $k > 1$ and even if $\mathbf{X}_{S^\star}$ is sufficiently "distinguishable": for instance, $\widetilde{\mathsf{AttHead}}_{\mathbf{K},\mathbf{V},\mathbf{q}}(\mathbf{h}_1,...,\mathbf{h}_L)$ is invariant to the order of $\mathbf{h}_1,...,\mathbf{h}_L$ (and thus of the tokens, in the typical case that $\mathbf{h}_i$ is an embedding of $\mathbf{X}_i$), whereas $g^\star(\mathbf{X}_{S^\star})$ may not be. Decoder-only transformers correct for this deficiency by (amongst other factors, such as positional embeddings) stacking multiple layers of attention heads. When doing so, embedding vectors $\mathbf{h}_i$ at individual positions of higher layers can contain information about *multiple* tokens of the input sequence. Inspired by this property, we introduce the following model of an "idealized transformer" – instead of stacking multiple attention heads, it directly applies attention on *subsets* of $k$ tokens:

**Definition 3.3** (Sparse functional attention class)**.** Fix a positive integer $k$ (the sparsity) together with hypothesis classes $\mathcal{G}^{\mathsf{key}}$ consisting of functions mapping $(\mathbb{N} \times \mathcal{V})^k \to (\mathbb{R} \cup \{-\infty\})$ and $\mathcal{G}^{\mathsf{val}}$ mapping $\mathcal{V}^k \to \mathcal{Y}$. We define the *$k$-sparse functional attention class* $\mathcal{H}^{\mathsf{attn}} = \mathcal{H}^{\mathsf{attn}}(\mathcal{G}^{\mathsf{key}}, \mathcal{G}^{\mathsf{val}})$ to be the class of all $h_{g_0,g_1} : \mathcal{V}^\star \to \mathcal{Y}$, indexed by $g_0 \in \mathcal{G}^{\mathsf{key}}, g_1 \in \mathcal{G}^{\mathsf{val}}$, where[4]

$$h_{g_0,g_1}(\mathbf{x}_1,...,\mathbf{x}_L) := \sum_{S \in \mathrm{Sets}_k([L])} \frac{\exp(g_0(S,\mathbf{x}_S)) \cdot g_1(\mathbf{x}_S)}{\sum_{S' \in \mathrm{Sets}_k([L])} \exp(g_0(S',\mathbf{x}_{S'}))}.$$

Intuitively, one should think of the "attention function" $g_0(S,\mathbf{x}_S)$ as computing attention scores based off of *sets* of $k$ tokens. Sparse functional attention with $k=1$ captures a single

---

[2]In the case of decoder-only transformers, $\mathbf{q}$ is commonly taken to be equal to be a linear function applied to $\mathbf{h}_L$, though in this discussion we take it to be fixed along with the other parameters of the attention head.

[3]Notice that since $S^\star$ is not given explicitly, there are certainly cases where it is information-theoretically impossible to determine $S^\star$ from $\mathbf{X}$. However, we give several examples in Appendices B.1 and C.3 where the planted sequence $\mathbf{X}_{S^\star}$ is sufficiently "distinguishable" from the remaining tokens $\mathbf{X}_i, i \notin S^\star$, and in fact $\mathbf{Y}$ can indeed be computed with high probability.

[4]If $g_0(S, \mathbf{x}_S) = -\infty$ for all $S \in \mathrm{Sets}_k([L])$, then we set $h_{g_0,g_1}(\mathbf{x}_1,...,\mathbf{x}_L)$ to be $\frac{1}{|\mathrm{Sets}_k([L])|} \sum_S g_1(\mathbf{x}_S)$.

---

attention head with fixed query vector as discussed above: in particular, for $S = \{i\}$, we choose $g_0(S,\mathbf{x}_S) = \langle \mathbf{q}, \mathbf{K}\mathbf{h}_i \rangle$ where $\mathbf{h}_i$ is the embedding of $\mathbf{x}_i$, and $g_1(\mathbf{x}_i) = \mathbf{V} \cdot \mathbf{h}_i$. The fact that $g_0(S,\mathbf{x}_S)$ can also depend on the position indices $S$ should be interpreted as allowing us to model positional embeddings (see Section 4.1 for more discussion). The following proposition formalizes these observations, verifying that the sparse group attention class indeed generalizes attention heads.

**Proposition 3.1** (Informal version of Proposition B.2)**.** *The class of attention heads corresponding to a vocabulary $\mathcal{V}$ and embedding dimension $d$ is equal to the 1-sparse functional attention class $\mathcal{H}^{\mathsf{attn}}(\mathcal{G}^{\mathsf{key}}, \mathcal{G}^{\mathsf{val}})$ for appropriate choices of $\mathcal{G}^{\mathsf{key}}, \mathcal{G}^{\mathsf{val}}$.*

**Additional properties.** We believe that the abstraction of sparse functional attention defined in Definition 3.3 may be of broader interest in obtaining a theoretical understanding of various properties of transformers. Nevertheless, in order to analyze length generalization, we need to further restrict the class $\mathcal{H}^{\mathsf{attn}}(\mathcal{G}^{\mathsf{key}}, \mathcal{G}^{\mathsf{val}})$ in the following ways: first, we assume that the class $\mathcal{G}^{\mathsf{key}}$ is *local*, meaning that it only outputs finite (i.e., not negative infinity) scores on subsets $S \subset \mathbb{N}$ for which $\max(S) - \min(S)$ is bounded. Second, we assume that $\mathcal{G}^{\mathsf{key}}$ only uses *relative positional information*, meaning that shifting $S$ does not change $g_0(S,\mathbf{x})$ for all $g_0 \in \mathcal{G}^{\mathsf{key}}, \mathbf{x} \in \mathcal{V}^k$.

**Assumption 3.2** ($\mathcal{G}^{\mathsf{key}}$ is local and relative)**.** We introduce the following properties of the class $\mathcal{G}^{\mathsf{key}}$:

1. Fix $L_{\mathsf{local}} \in \mathbb{N}$. We say that $\mathcal{G}^{\mathsf{key}}$ is $L_{\mathsf{local}}$-*local* if for all $S \in \mathrm{Sets}_k(\mathbb{N})$ for which $\max_{i \in S} i - \min_{i \in S} i > L_{\mathsf{local}}$, we have $g_0(S,\mathbf{x}) = -\infty$ for all $g_0 \in \mathcal{G}^{\mathsf{key}}, \mathbf{x} \in \mathcal{V}^k$.

2. We say that $\mathcal{G}^{\mathsf{key}}$ is *relative* if for any $i \in \mathbb{N}$ and $S \in \mathrm{Sets}_k(\mathbb{N})$, $\mathbf{x} \in \mathcal{V}^k$, it holds that $g_0(S,\mathbf{x}) = g_0(S+i,\mathbf{x})$ for all $g_0 \in \mathcal{G}^{\mathsf{key}}$.[5]

The assumption that $\mathcal{G}^{\mathsf{key}}$ is relative is inspired by the fact that many common positional embeddings only use *relative* positional information. While the assumption of locality is strong, we remark that it is provably necessary (Appendix C.3) and can be relaxed using techniques such as positional coupling (Section 4.1). Throughout the paper, we will assume that the parameters $k, L_{\mathsf{local}}$ are chosen so that $L_{\mathsf{local}} \geq k$.

## 4. Theoretical results

In this section, we establish formal length generalization guarantees for sparse functional attention classes $\mathcal{H}^{\mathsf{attn}}$ (Definition 3.3) for distribution ensembles $\mathcal{P}$ with sparse planted correlation (Definition 3.2). To do so, we need to make a few assumptions on the ensemble $\mathcal{P}$. The first assumption ensures that the ensemble $\mathcal{P}$ is approximately $\mathcal{H}^{\mathsf{attn}}$-realizable.

**Assumption 4.1** (Realizability)**.** We assume that $\mathcal{P}$ is $\delta$-approximately $\mathcal{H}^{\mathsf{attn}}(\mathcal{G}^{\mathsf{key}}, \mathcal{G}^{\mathsf{val}})$-realizable, i.e., there is $h^\star \in \mathcal{H}^{\mathsf{attn}}(\mathcal{G}^{\mathsf{key}}, \mathcal{G}^{\mathsf{val}})$ so that $\mathbb{E}_{(\mathbf{X},\mathbf{Y}) \sim \mathcal{P}_\ell}[\mathcal{L}(h^\star(\mathbf{X}),\mathbf{Y})] \leq \delta$ for all $\ell$.

---

[5]For $S = \{i_1,...,i_k\}$, $S+i := \{i_1+i,...,i_k+i\}$.

Our second assumption states that the distributions $\mathcal{Q}_\ell^{\text{pos}}$ in the context of Definition 3.2 have bounded coverage at all locations in the sense that changing $\ell$ or shifting $S$ by $\Delta$ units does not significantly change $\mathcal{Q}_\ell^{\text{pos}}(S)$.

**Assumption 4.2** (Coverage of $\mathcal{Q}_\ell^{\text{pos}}$)**.** Fix a $k$-sparse distribution ensemble $\mathcal{P}$ specified by $\mu, (\mathcal{Q}_\ell^{\text{pos}})_{\ell \in \mathbb{N}}, \mathcal{Q}^{\text{voc}}$, as well as $L_{\text{local}} \in \mathbb{N}$. We assume that for each $\ell \in \mathbb{N}$ with $\ell \geq L_{\text{local}}$, there is some positive value $\eta_\ell \in \mathbb{R}$ so that for all $\ell' \in [L_{\text{local}}, \ell]$, $\Delta \geq 0$ and $S^\star \in \text{Sets}_k([\ell])$, $\frac{\mathcal{Q}_\ell^{\text{pos}}(S^\star)}{\mathcal{Q}_{\ell'}^{\text{pos}}(S^\star - \Delta)} \leq \eta_\ell$ and $\frac{\mathcal{Q}_{\ell'}^{\text{pos}}(S^\star)}{\mathcal{Q}_\ell^{\text{pos}}(S^\star)} \leq \eta_\ell$, where the first inequality is only required if $S^\star - \Delta \in \text{Sets}_k([\ell'])$ and the second inequality is only required if $S^\star \in \text{Sets}_k([\ell'])$.

Our main result states that for any classes $\mathcal{G}^{\text{key}}, \mathcal{G}^{\text{val}}$ and distribution ensemble $\mathcal{P}$ satisfying Assumptions 3.2, 4.1 and 4.2, the class $\mathcal{H}^{\text{attn}}(\mathcal{G}^{\text{key}}, \mathcal{G}^{\text{val}})$ enjoys length generalization with respect to the ensemble $\mathcal{P}$.

**Theorem 4.3** (Provable length generalization)**.** *Fix any* $k \in \mathbb{N}$*, and consider any $k$-sparse functional attention class* $\mathcal{H}^{\text{attn}} = \mathcal{H}^{\text{attn}}(\mathcal{G}^{\text{key}}, \mathcal{G}^{\text{val}})$ *which is $L_{\text{local}}$-local and relative (Assumption 3.2) for some $L_{\text{local}} \in \mathbb{N}$. Then for any $k$-sparse planted correlations distribution ensemble $\mathcal{P}$ (Definition 3.2) satisfying Assumptions 4.1 and 4.2, and any integers $L, \bar{L}$ for which $L_{\text{local}} \mid \bar{L} - L$ and $L \geq 4L_{\text{local}}$, $\mathcal{H}^{\text{attn}}$ achieves* $(L, \bar{L}, \eta_L \eta_{\bar{L}} \cdot L \cdot (\bar{L}/L_{\text{local}})^2 \cdot \delta)$-*length generalization with respect to the ensemble $\mathcal{P}$.*

In Appendix C.3, we show that if we remove any one of the main assumptions of Theorem 4.3, then length generalization can fail to hold. Moreover, we remark that the error $\eta_L \cdot \eta_{\bar{L}} \cdot L \bar{L}^2 \cdot \delta$ on distributions of length $\bar{L}$ can be improved; see Appendix C.2 for further discussion.

### 4.1. Improving length generalization: positional coupling

As discussed above, one limitation of Theorem 4.3 is its reliance on Item 1 of Assumption 3.2, which is not satisfied in actual transformers: indeed, transformers may attend to tokens which are very far apart, as such dependencies can occur in natural language. Is there a way to mitigate this limitation?

We now show how a theoretical abstraction of position coupling as discussed in Section 2.1 can allow us to remove this locality requirement. Roughly speaking, this abstraction of position coupling states that there is a joint distribution over $(S^\star, \psi_\ell)$, where $\psi_\ell : [\ell] \to [\ell]$ gives a way of "rewriting" position indices, so that the "rewritten" set $\psi_\ell(S^\star)$ of indices in the planted set $S^\star$ satisfies the locality condition of Assumption 3.2. In particular, for each position ID $i \in [\ell]$, the value of $\psi_\ell(i)$ should be interpreted as its "coupled position ID" as discussed in Section 2.1. More precisely, we have (see Appendix D for the full definition):

**Definition 4.1** (Informal version of Definition D.2)**.** Fix a distribution ensemble $\mathcal{P}$ with $k$-sparse planted correlations per Definition 3.2 (defined by $\mu, \mathcal{Q}^{\text{pos}}, \mathcal{Q}^{\text{voc}}$). A $L_{\text{local}}$-local *position coupling* of $\mathcal{P}$ is defined by, for each $\ell \in \mathbb{N}$, a joint dis-

tribution $\mathcal{Q}^{\text{pos-c}}$ over tuples $(S^\star, \psi_\ell)$ with $S^\star \in \text{Sets}_k([\ell])$ and $\psi_\ell : [\ell] \to [\ell]$ so that the marginal of $S^\star$ under $\mathcal{Q}^{\text{pos-c}}$ is $\mathcal{Q}_\ell^{\text{pos}}$ and with probability 1 under the draw of $(S^\star, \psi_\ell) \sim \mathcal{Q}^{\text{pos-c}}$:

1. $\max\{\psi_\ell(S^\star)\} - \min\{\psi_\ell(S^\star)\} \leq L_{\text{local}}$.

2. $\psi_\ell, S^\star$ satisfy some additional conditions which, e.g., prevent coupling of non-planted tokens (i.e., outside of $S^\star$; Definition D.2).

With Definition D.2 in hand, we can now describe how hypotheses in a sparse functional attention class $\mathcal{H}^{\text{attn}}$, which *may not satisfy locality*, can be transformed into new hypotheses which will use information from the output of the "position coupling" $\psi_\ell$ and *will also satisfy locality*. Roughly speaking, given a sample $(\mathbf{X}, S^\star, \mathbf{Y}) \sim \mathcal{P}_\ell$, we will join (i.e., "couple") all tokens $\mathbf{X}_i$ for which $\psi_\ell(i)$ is identical. We will denote the resulting distribution over sequences of "coupled tokens" by $\text{PC}[\mathcal{P}_\ell]$. Moreover, we define a "position-coupled" hypothesis class $\text{PC}[\mathcal{H}^{\text{attn}}]$ whose hypotheses, when given sequences of coupled tokens as above, "unpacks" them and applies the corresponding hypothesis in $\mathcal{H}^{\text{attn}}$. In Appendix D, we formally define these notions and prove the following proposition, which shows that this procedure of position coupling can remove the locality requirement from Theorem 4.3:

**Proposition 4.4** (Informal version of Proposition D.2)**.** *If $\mathcal{H}^{\text{attn}}$ is a sparse functional attention class and $\mathcal{P}$ is an ensemble with $k$-sparse planted correlations which satisfies $\delta$-approximate strong realizability (Assumption D.1) and Assumption 4.2, then for $L, \bar{L}$ satisfying the conditions of Theorem 4.3, $\text{PC}[\mathcal{H}^{\text{attn}}]$ achieves* $(L, \bar{L}, \eta_L \eta_{\bar{L}} \cdot L\bar{L}^2 \cdot \delta)$-*length generalization with respect to the ensemble* $\text{PC}[\mathcal{P}]$.

## 5. Experiments

Conceptually, we view the main takeaways of Theorem 4.3 and Proposition 4.4 to be the following:

(T1) First, an important factor enabling length generalization is that the label $\mathbf{Y}$ depends on only $k$ tokens of the input (in the sense of Definition 3.2 and in particular the relation $\mathbf{Y} = g^\star(\mathbf{X}_{S^\star})$ in (2)). In particular, the parameter $k$ (which we refer to informally as the *sparsity*) must be the same for both the lengths on which we train (namely, lengths $\ell \leq L$) and the length $\bar{L}$ to which we attempt to extrapolate, and sufficiently small compared to the maximum training length $L$.

(T2) Second, *locality* of the hypothesis class (per Item 1 of Assumption 3.2) plays an important role as well: the maximum distance $L_{\text{local}}$ between tokens which "matter" in predicting the label $\mathbf{Y}$ must be the same for lengths $\ell \leq L$ on which we train and the length $\bar{L}$ to which we extrapolate. Moreover, as this requirement is quite strong (and unrealistic for many problems of interest), one way to mitigate it is the technique of position coupling (per Proposition 4.4).

In this section, we evaluate these conclusions for synthetic and

natural language modeling data.

## 5.1. Length generalization for sparse parity

First, we discuss a simple setting which measures the degree to which length generalization in transformers reflects the requirement from (T1) that the sparsity $k$ be sufficiently small and not grow as a function of the sequence's length. In particular, consider the following *sparse parity* task:[6] given $k, \ell \in \mathbb{N}$, the input is drawn from a distribution $\mathcal{D}_{k,\ell}^{\mathsf{sp}}$ over length-$2\ell$ sequences, where tokens at even-numbered positions are bits, and tokens at odd-numbered positions belong to some set $\Omega$. Exactly $k$ tokens at odd-numbered positions belong to some "special subset" $\Omega' \subset \Omega$, and the goal is to find the parity of the $k$ tokens immediately following them. See Appendix E.1.1 for precise details.

**Experimental setup: data.** For each value of $K_{\mathsf{train}} \in \{4,6,8,10,12\}$, we train a transformer $\hat{h}_{K_{\mathsf{train}}}$ to predict the last token of samples drawn from $\mathcal{D}_{\ell,k}^{\mathsf{sp}}$, where $\ell$ is sampled uniformly subject to the length of the sample satisfying $2\ell \in [20, 50]$ and $k \sim \mathrm{Unif}([K_{\mathsf{train}}])$. We then evaluate the performance of each of the trained transformers $\hat{h}_{K_{\mathsf{train}}}$ on samples drawn from $\mathcal{D}_{\ell, k_{\mathsf{test}}}$ of length $2\ell \in [20, 500]$ and with sparsities $k_{\mathsf{test}} \in \{4,6,8,10,12,14,16\}$.

**Experimental setup: model.** Our model is based off of the GPT-NeoX (decoder-only) transformer (Andonian et al., 2023), and uses rotary positional embeddings (RoPE). To ensure nontrivial length generalization performance, we combined RoPE with PoSE (Appendix B.3).[7] Full training and evaluation details may be found in Appendix E.

**Remark 5.1.** Numerous other modifications to position IDs have been proposed for length generalization, such as *position interpolation* and various enhancements, which typically modify the way the transformer uses the position IDs at *inference time*, often after a small amount of fine-tuning (see Appendix A.2). We stick with PoSE in this paper (when position coupling is not applicable) because of its simplicity and since (a) it does not require modifying the transformer's computations at inference time; and (b) the fine-tuning for position interpolation requires sequences of length given by the *testing* context length, which fails to lie in our framework where we assume that any amount of training on such sequences is not allowed. Understanding the role of sparsity and locality for length generalization in transformers which make these inference-time modifications to position IDs is left for future work.

[6]This task is slightly different from standard formulations of sparse parity, where the sparsely chosen positions are not identified as part of the input. We use this version so as to allow a different set of $k$ positions to be chosen for each example.

[7]This task is not well-suited for position coupling, since the output token depends on $2k$ input tokens, all of which have different position IDs.

**Results & discussion.** The accuracy of the trained transformers $\hat{h}_{K_{\mathsf{train}}}$ for predicting the last (parity) token on samples of various lengths $\ell$ and sparsity values $k$ is shown in Figure 1a for $K_{\mathsf{train}} = 10$ and in Figure 3 for the remaining values of $K_{\mathsf{train}}$. Two observations are notable: first, when the training sparsity $K_{\mathsf{train}}$ is small enough (i.e., $K_{\mathsf{train}} \leq 8$; see Figure 3), then $\hat{h}_{K_{\mathsf{train}}}$ experiences near-perfect length generalization up to lengths of 500, for all values of the test sparsity $k_{\mathsf{test}} \leq K_{\mathsf{train}}$. However, for test sparsity values $k_{\mathsf{test}} > K_{\mathsf{train}}$, the performance of $\hat{h}_{K_{\mathsf{train}}}$ deteriorates rapidly, for both in-distribution and out-of-distribution lengths. This behavior is consistent with Theorem 4.3, which implies that good length generalization occurs as long as the sparsity does not change between the train and test distributions and is sufficiently small as a function of the training length.

A corollary of this reasoning is that for a fixed maximum training length $L$, the length generalization behavior with respect to the distribution ensemble $(\mathcal{D}_{\ell,k}^{\mathsf{sp}})_{\ell \in \mathbb{N}}$ should degrade at some point if the sparsity $k$ is allowed to increase enough. (In particular, our theoretical guarantee on length generalization only holds for sufficiently small training sparsity, i.e., $K_{\mathsf{train}} \leq L/4$ in the case of Theorem 4.3, though the particular threshold is likely different in the case of actual transformers). This behavior is mirrored in Figure 1a as well as Figure 3 in Appendix E.1.2, where, for each fixed value of $K_{\mathsf{train}} \in \{10, 12\}$, larger values of $k \in [K_{\mathsf{train}}]$ have worse length generalization despite all having near-perfect in-distribution performance.[8]

## 5.2. Scratchpad with Predictive Position Coupling

Many tasks, unlike sparse parity, have the property that the sparsity of the next-token prediction task (as formalized by, e.g., Definition 3.2) can be quite large and in fact *increase* from shorter to longer lengths. To help achieve length generalization, one approach which has been suggested in numerous existing works (Anil et al., 2022; Hou et al., 2024) is the *chain-of-thought/scratchpad* technique (Nye et al., 2022; Wei et al., 2022; Lewkowycz et al., 2022). It proceeds by computing tokens representing intermediate steps, such as intermediate parities when the task is to compute the parity of a sequence of $\ell$ bits. It has been observed that the scratchpad technique alone is insufficient to ensure length generalization (Anil et al., 2022; Dziri et al., 2023; Hu et al., 2024; Kazemnejad et al., 2023; Lanchantin et al., 2023). However, a number of recent works (Cho et al., 2024a;b; McLeish et al., 2024) have shown that the technique of *position coupling* (see Section 2.1), when combined with a scratchpad (and even when used on its own, when appropriate), can allow significant length generalization on arithmetic tasks such as parity, addition, and multiplication. This development parallels our theoretical results in Section 4.1, where

[8]Recall that $\hat{h}_{K_{\mathsf{train}}}$ was trained by placing equal weights on all sparsities $k \leq K_{\mathsf{train}}$, so in this respect the distributions $\mathcal{D}_{\ell,k}^{\mathsf{sp}}$ for $k \leq K_{\mathsf{train}}$ were treated "on equal footing."

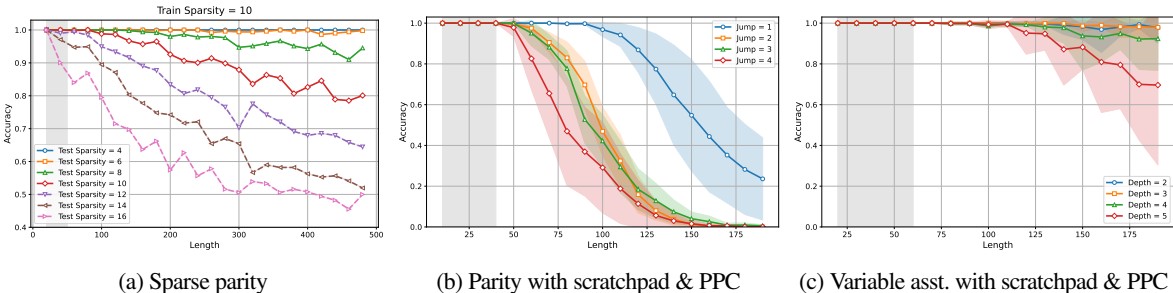

(a) Sparse parity     (b) Parity with scratchpad & PPC     (c) Variable asst. with scratchpad & PPC

Figure 1: Length generalization for experiments in Sections 5.1 and 5.2. Shaded areas in (b), (c) represent 95% CIs over multiple training runs.

we showed that position coupling removes the stringent requirement of locality, which is not satisfied in practical settings.

One downside of position coupling is that it requires significant structure in the format of the data sequence, including the scratchpad: in particular, once the length is fixed, the *coupled position IDs* (see Section 2.1) in all tokens the transformer is trained to predict must be determined, as they must be fed into the model to predict each successive token. (For instance, for the string reversal example in Section 2.1, when $L$ denotes the length of the string to be reversed, the sequence of position IDs is given as in Section 2.1.) As a consequence, the applications of position coupling are currently limited to a small number of synthetic tasks where the computation of each token in the sequence (including the scratchpad) can be coupled with a fixed position of the input.

Towards relaxing this limitation to allow sequences where the coupled position IDs can be *input-dependent*, we propose *Predictive Position Coupling* (PPC). PPC functions as follows:

1. It modifies the transformer architecture to instead *predict* the coupled position ID for each next token. In particular, we add an additional output embedding module so that, at each step, the transformer predicts two IDs: the next token ID, as well as the coupled position ID for that token.

2. The transformer is trained to predict both the actual tokens, as well as the coupled position IDs for each token. We use the standard cross-entropy loss for both predictions, and weight them equally.

3. At generation time, the two predicted tokens at each step are fed in as the position ID and token ID at the following position.

In our results, we report the fraction of examples on which the model correctly predicts *all tokens and coupled position IDs*.

**Experimental setup: model.** We use the same NeoX model as described in Section 5.1, with the following modifications to implement position coupling. Following (Cho et al., 2024a;b; McLeish et al., 2024), we use absolute position embeddings and fix some integer $L_{\mathsf{max}}$ denoting the maximum possible length of test-time sequences. During training, we shift each

sequence of coupled position IDs by a uniformly random offset subject to the constraint that all shifted position IDs be at most $L_{\mathsf{max}}$. In particular, if the sequence of coupled position IDs ranges from 1 to $\ell$, then we would shift by a uniform element of $\{0,1,...,L_{\mathsf{max}}-\ell\}$. We remark that using RoPE with PoSE (as opposed to PPC) performs significantly worse; see Appendix E.

5.2.1. WARMUP: PARITY WITH SCRATCHPAD.

As a warmup, we first consider the task of evaluating the parity of a sequence of $\ell$ bits with use of a scratchpad (and PPC) to compute intermediate parities (as in, e.g., Anil et al. (2022); Cho et al. (2024b)). The scratchpad is structured as follows: it has length $\ell$, and the $i$th token of it is the parity of the first $i$ tokens of the input; thus, the final ($\ell$th) token of the scratchpad is the desired answer, i.e., the parity of all $\ell$ input tokens. To measure the impact of sparsity, we consider the following modification of the standard scratchpad format, which we refer to as having a scratchpad with "jumps". For values of $k \in \{1,2,3,4\}$, we only write every $k$th bit in the scratchpad, meaning that the $i$th bit of the scratchpad is the parity of the first $ki$ tokens of the input. In particular, the case $k=1$ corresponds to the standard scratchpad format. We also modify the position coupling IDs so that each position ID is repeated $k$ times in the input sequence. Full details of the data and position coupling formats are in Appendix E.2.1.

**Results & discussion.** Figure 1b shows the length generalization behavior for our transformer model trained as discussed above, for "jump lengths" $k \in \{1,2,3,4\}$. Note that the jump length $k$ parametrizes the sparsity of the next-token prediction task: each token in the scratchpad depends on $k+1$ previous tokens (namely, the immediately preceding one together with the $k$ tokens in the input which correspond to that step of computation). While all 4 jump lengths attain near-perfect accuracy for in-distribution lengths, smaller values of $k$ have superior performance on longer out-of-distribution lengths, consistent with our takeaway (T1). This fact is particularly notable in light of the fact that for jump length $k$, the scratchpad is *shorter* by a factor of $k$, which a priori could presumably have made it easier to correctly predict all tokens and position IDs correctly (the metric measured in Figure 1b). Thus, at least in this task, *spar-*

*sity (as measured here by $k$) is a more important determinant of length generalization than the length of the output sequence.*

### 5.2.2. VARIABLE ASSIGNMENT WITH SCRATCHPAD.

Next, we consider the more challenging task of *variable assignment* with a scratchpad, of which several slight variants have been studied in numerous prior works (Zhang et al., 2023; 2022; Lanchantin et al., 2023; Peng et al., 2024).

**Experimental setup: data overview.** In the variable assignment problem, we fix a *depth* $d$, denoting the number of "hops" it takes to compute the final value. Roughly speaking, the goal is as follows: given a sequence of "variable assignments" of the form $v \leftarrow v'$ (where $v, v'$ are token IDs), together with a *starting variable* $v_{\text{init}}$, follow a sequence of $d$ assignments starting at $v_{\text{init}}$. In more detail, a problem instance consists of some number of "chains", each of the form $v_1 \leftarrow v_2; v_2 \leftarrow v_3; \cdots; v_d \leftarrow v_{d+1}$ for some *depth parameter* $d$. The variable assignments in each of these chains are interleaved randomly; exactly one of the chains has $v_1 = v_{\text{init}}$. The goal of the task is to find the final variable $v_d$ on the chain for which $v_1 = v_{\text{init}}$. To do so, we use a scratchpad which writes out in full the chain starting with $v_{\text{init}}$, together with position coupling, which matches each element of this chain on the scratchpad with its corresponding position in the input. The data format and position coupling scheme are presented in detail in Appendix E.3. We mention here that, because the chains are interleaved randomly, the position IDs used in the position coupling scheme *depend on the particular input instance*. Thus, the use of Predictive Position Coupling is crucial in order to be able to feed in the correct coupled position IDs in the scratchpad at inference time.[9] We train on input sequences of length $\ell \in [20, 60]$, and test on lengths $\bar{L} \in [20, 200]$.

**Results.** Figure 1c shows our results for the variable assignment problem with position coupling: the technique of Predictive Position Coupling allows good length generalization for test lengths up to 3 times the training sequence length, for depths $d \in \{2, 3, 4, 5\}$. In Figs. 5a to 5c (Appendix E.3.2), we present baselines that remove either just the position coupling or the scratchpad altogether; as predicted by takeaways (T2) and (T1), respectively, these modifications significantly harm length generalization. *In particular, using RoPE with PoSE (with or without scratchpad) performs significantly worse than PPC.*

### 5.3. Sparsity & length generalization in natural language

Finally, we study the interplay between sparsity and length generalization in natural language. A trivial way to obtain generalization from sequences of length $\leq L$ to those of length $\bar{L} > L$ is to simply ignore tokens more than $L$ units in the past.

To understand length generalization in models beyond this baseline, we ask: *given $L < \bar{L}$, can a transformer trained on natural language sequences of length $L$ achieve smaller perplexity on sequences of length $\bar{L}$ than on sequences of length $L$?* Previous works have shown a positive answer to this question (as will our experiments) when one uses appropriate modifications to the positional embeddings, namely PoSE (Zhu et al., 2023) or position interpolation (Chen et al., 2023; emozilla, 2023; Peng et al., 2023; Chen et al., 2024). In light of this, a natural way to measure sparsity is as follows: *is there a way to select a small number $k$ of the tokens more than $L$ units in the past from the present token, so that, using* just *those $k$ tokens together with the most recent $L$ tokens, we can nearly recover the perplexity of the model on* full *sequences of length $\bar{L}$?* Informally, we are asking if the dependence on tokens more than $L$ units in the past is *sparse*, in terms of beating the trivial baseline for length generalization.

**Experimental setup.** We trained our models using the OLMo codebase (Groeneveld et al., 2024) on the C4 dataset (Raffel et al., 2019). We trained a transformer model $\hat{h}_{\text{short}}$ using context length $L = 64$.[10] Our aim is to length extrapolate to a context length of $\bar{L} = 2L = 128$. We used the PoSE technique with $L_{\text{max}} = \bar{L}$ and $c = 2$ chunks. We found that training from scratch using PoSE did not yield good performance at length $\bar{L}$, so instead we first trained the model without PoSE for roughly 70% of the iterations and on the final 30% of the iterations used PoSE. Precise hyperparameters may be found in Appendix F.

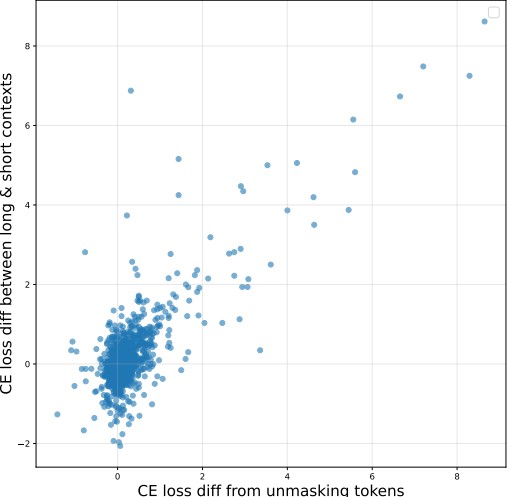

Figure 2: Cross-entropy loss differences obtained by unmasking $k$ influential tokens ($x$-axis) or all of the first 64 tokens ($y$-axis).

**Results & discussion.** Table 5 confirms that the perplexity of $\hat{h}_{\text{short}}$ on a context window of 128 is less than that on a context window of 64, verifying that $\hat{h}_{\text{short}}$ is able to utilize information

---

[9]Simply feeding in the correct value as the next position ID would be "cheating" as it would be performing part of the relevant computation of the scratchpad.

[10]We chose the context length to be so short in order to decrease the performance of the trivial baseline for length generalization which ignores all but the $L$ most recent tokens.

beyond the shorter length-64 training context window. Moreover, in Figure 2, we plot, for each of a total of $500$ evaluation data points $\mathbf{X}_{1:\bar{L}}$ with context window $\bar{L}=128$, the amount of decrease in cross-entropy loss for predicting the *last* token $\mathbf{X}_{\bar{L}}$ of the sequence one achieves by the following two strategies:

(a) Unmasking $k=5$ "influential" tokens at positions $i \leq L = 64$ compared to masking out all tokens at positions $i \leq 64$ (this difference is the $x$-coordinate);

(b) Unmasking *all* tokens at positions $i \leq 64$ compared to masking out all tokens at positions $i \leq 64$ (this difference is the $y$-coordinate).

In particular, the $y$-coordinate of each point should be interpreted as the amount the model uses the first $64$ tokens to improve the prediction of the last token, and the $x$-coordinate of each point should be interpreted as the amount the model uses a select $k$ of the first $64$ coordinates (see Figure 6 for an illustration). As can be seen, the data points are roughly clustered along the line $x=y$, which indicates that (up to noise) the model's prediction of the last token in the sequence is sparse in its dependence on tokens "far in the past", consistent with Section 5 which predicts that such sparsity allows $\hat{h}_{\text{short}}$ to length-generalize. We discuss additional details and experiments (including how the $k$ influentail tokens are chosen) in Appendix F.

## Acknowledgements

NG was supported in part by a Fannie & John Hertz Foundation Fellowship and an NSF Graduate Fellowship. We thank anonymous reviewers for very helpful feedback.

## Impact Statement

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

# A. Related work

## A.1. Theoretical guarantees for length generalization

Zhou et al. (2023); Huang et al. (2024) propose that transformers will length-generalize at tasks which can be solved using a short RASP program (Weiss et al., 2021). Theorem 7 of Huang et al. (2024), however, assumes that the transformer is chosen so as to minimize a certain regularizer tailored specifically for length generalization, and their result is asymptotic in nature (in contrast to ours). Sabbaghi et al. (2024) show that using gradient flow on a 1-layer linear transformer model with relative position embeddings on a simple linear regression task will converge to a model which length-generalizes, whereas the use of absolute position embeddings will fail to length-generalize. Ahuja & Mansouri (2024) show that a model class defined as a simple abstraction of attention heads that sums up pairwise dependencies of tokens experiences length generalization. Unlike our theoretical results, those of Sabbaghi et al. (2024); Ahuja & Mansouri (2024) are asymptotic in nature and cannot capture the (most common) case of a *softmax* attention head. Moreover, they proceed, roughly speaking, by showing that the learned model is equal to the ground-truth model *on the entire domain*, which is generally *not* the case with actual language models (Zou et al., 2023; Wei et al., 2023; Andriushchenko et al., 2024). As our theoretical setup incorporates distributional assumptions, it establishes length generalization in cases where the learned model is only accurate on in-distribution inputs, which we believe to be more realistic. Moreover, all of the preceding works generally apply only to specific classes of *transformers or linear classes*; in contrast, our framework, while being able to capture an attention head, is significantly more general in that we allow arbitrary, potentially nonlinear, function classes (see Definition 3.3), thus meaning our results may have significance for other (non-transformer) classes of models as well.

Hou et al. (2024) proposes to use *Turing programs*, a scratchpad strategy inspired by Turing machines, to achieve length generalization on an array of tasks. Their theoretical results, however, are only representational in nature, showing that transformers can represent Turing programs without accounting for what models algorithms such as risk minimization will actually learn. Hahn & Rofin (2024) offer reasons why transformers struggle to length-generalize on parity, based on their sensitivity bias.

Further from our own work, Marsden et al. (2024) obtain provable guarantees for length generalization in the context of dynamical systems. Wang et al. (2024) prove that gradient descent on transformers can provably learn a task known as the *sparse token selection task* (Sanford et al., 2023), which bears resemblance to Definition 3.2 (indeed, a slight modification of the task defined in Section 2.1 of Wang et al. (2024) is in fact a special case of a distribution with sparse planted correlations (Definition 3.2)).

## A.2. Modifying positional embeddings for length generalization

In addition to position coupling (Cho et al., 2024a;b) (and the closely related Abacus embeddings (McLeish et al., 2024)), which is a focus in our paper, several other techniques have been developed to modify position embeddings during training and/or inference time to improve the length generalization performance of transformers. Zhu et al. (2023) developed the positional skip-wise technique (PoSE; see Appendix B.3), which was later refined by modifying the distribution of position IDs used at training time in Wu et al. (2024). We remark that PoSE is conceptually similar to *randomized position embeddings* (Ruoss et al., 2023), which trains using a random set of position IDs from the test-length context window (with no guarantee on the contiguity of these IDs, unlike PoSE).

Another popular strategy to extend the context length of transformers which has seen traction for models at larger scales, such as as Code Llama (Rozière et al., 2024), is *position interpolation*. This technique *scales down* ("interpolates") the position IDs in the longer test-length context window to match the length of the shorter training-length context. Several such interpolation strategies have been proposed, including the canonical choice of *linear interpolation* (Chen et al., 2023), as well as *NTK-RoPE* (emozilla, 2023), *YaRN* (Peng et al., 2023), and *CLEX* (Chen et al., 2024); the latter strategies adjust the amount of interpolation done on a per-frequency basis, with different RoPE frequencies receiving different interpolation scales. One downside of these position interpolation strategies is that they generally require some fine-tuning on the longer test-length sequences in order to effectively use the longer context windows (e.g., to achieve decreased perplexity on longer sequences than those seen during training). Such fine-tuning complicates the theoretical setting of length generalization, where it is typically assumed that any amount of training on the longer test-length sequences is not allowed. We remark, however, that these position interpolation techniques can be combined with PoSE (Zhu et al., 2023; Wu et al., 2024); exploring such combinations in the context of our experiments is an interesting direction for future work.

Finally, we remark that some strategies, such as LM-Infinite (Han et al., 2024) and Self-Extend (Jin et al., 2024), have been proposed to adjust the attention mechanism at inference time so as to achieve length generalization without any fine-tuning, though their performance lags somewhat (Lu et al., 2024).

### A.3. Empirical evaluations & explanations of length generalization

A number of papers have offered empirical evaluations and comparisons of various techniques for length generalization. Anil et al. (2022) studied a few simple length generalization tasks, such as parity and variable assignment, and found that techniques such as finetuning and using a scratchpad did not lead to much length generalization. Kazemnejad et al. (2023) compared the performance of various positional encoding techniques for length generalization, and found that NoPE performed best (though their analysis did not account for techniques such as position interpolation and PoSE which can significantly improve length generalization for, e.g., RoPE). Lee et al. (2024) observed that length generalization is difficult for transformers on arithmetic tasks. Finally, Lu et al. (2024) performed a systematic study comparing various approaches to extend the context length of LLMs, including various types of position interpolation.

Ontanon et al. (2022); Dziri et al. (2023); Hupkes et al. (2019) study the out-of-distribution performance of transformers by focusing on *compositional generalization*, which refers to the ability of transformers to compose individual tasks found in the training data. There is also extensive work more broadly on out-of-distribution generalization (Nagarajan et al., 2020; Abbe et al., 2023; Kalavasis et al., 2024).

Finally, a number of works (e.g., Hupkes et al. (2019); Liu et al. (2023); Delétang et al. (2022); Zhang et al. (2023); Hsieh et al. (2024)) introduce new benchmarks and datasets for studying length generalization and more broadly the performance of LLMs on long contexts.

**Additional techniques for length generalization.**  Many papers have introduced new types of positional encoding schemes with the hope of improved length generalization, including relative positional embeddings (Shaw et al., 2018; Dai et al., 2019; Huang et al., 2020; Ke et al., 2021), ALiBi (Press et al., 2021), Hard-Alibi (Jelassi et al., 2024), and FIRE (Li et al., 2024).

Many other types of techniques have been proposed, such as priming the training dataset with a few long sequences (Jelassi et al., 2023), modifying the format of the input (Shen et al., 2023; Hu et al., 2024) such as by using a scratchpad (Lanchantin et al., 2023), architectural changes (Csordás et al., 2021b; Fan et al., 2024), and combining several such techniques (Csordás et al., 2021a; Liu et al., 2022; Zhou et al., 2024). Moreover, it was recently shown that length generalization can be achieved via a *self-improvement* framework (Lee et al., 2025), which trains the model on sequences of increasing length, which are labeled by previous versions of the model. We remark that such techniques lie outside the scope of this paper, as we require that the context length be bounded above by $L$ throughout training. Moreover, such techniques still may suffer from the computational issues that plague longer context lengths.

## B. Additional Preliminaries

### B.1. Additional context on distributional assumptions

The following proposition rules out length generalization in the absence of distributional assumptions.

**Proposition B.1.** *Fix any $L, \bar{L} \in \mathbb{N}$ with $L < \bar{L}$, and a hypothesis class $\mathcal{H} \subset \mathcal{Y}^{\mathcal{V}^\star}$. Let $\mathcal{P}_1, ..., \mathcal{P}_L$ be $\mathcal{H}$-realizable distributions, realized by $h^\star$. Suppose that $\epsilon > 0$ and $\hat{h}$ defined in (1) satisfies $\sup_{\mathbf{X} \in \mathcal{V}^{\bar{L}}} \mathcal{L}(h^\star(\mathbf{X}), \hat{h}(\mathbf{X})) > \epsilon$. Then there is an ensemble $\mathcal{P}$ extending $\mathcal{P}_1, ..., \mathcal{P}_L$ so that $\mathcal{H}$ does not have $(L, \bar{L}, \epsilon)$-length generalization with respect to $\mathcal{P}$.*

Below, we give a simple example of a distribution ensemble having $k$-sparse planted correlations (per Definition 3.2).

**Simple example: $k$-gram retrieval.**  Suppose that $\mathcal{V} = [N]$ for some $N \in \mathbb{N}$, and we want to model the "$k$-gram retrieval" task of outputting the token following a certain $k$-gram in the sequence $\mathbf{X}$, which is closely related to the notion of *induction heads* (Jelassi et al., 2024; Olsson et al., 2022). Formally, for a fixed length $\ell$, the tuple $(\mathbf{X}, \mathbf{Y}) \in (\mathcal{V}^\ell, \mathcal{Y})$ is drawn as follows: first, we draw $\mathbf{X}_1, ..., \mathbf{X}_{\ell-k} \sim \mathrm{Unif}(\mathcal{V})^{\ell-k}$ independently, then choose a random $\ell' \sim \mathrm{Unif}([\ell - 2k - 1])$, and set $\mathbf{X}_{\ell-k+1:\ell} := \mathbf{X}_{\ell'+1:\ell'+k}$, and $\mathbf{Y} := \phi(\mathbf{X}_{\ell'+k+1})$ where $\phi : \mathcal{V} \to \mathcal{Y}$ is some embedding function. In particular, the tokens $\mathbf{X}_{\ell-k+1:\ell}$ specify the particular $k$-gram we want to query. This distribution falls into the framework of Definition 3.2 by letting $\mu = \mathrm{Unif}(\mathcal{V})$, letting $S^\star$ be uniform over length-$(2k+1)$ sequences $(\ell'+1, ..., \ell'+k+1, \ell-k+1, ..., \ell)$ where $\ell' \sim \mathrm{Unif}([\ell - 2k - 1])$, and $\mathcal{Q}^{\mathsf{voc}}$ be uniform over sequences $(\mathbf{Z}_1, ..., \mathbf{Z}_k, \mathbf{Z}_{k+1}, \mathbf{Z}_1, ..., \mathbf{Z}_k)$ where $\mathbf{Z}_1, ..., \mathbf{Z}_{k+1} \sim \mathrm{Unif}(\mathcal{V})$. Finally, $g^\star(\mathbf{Z}_1, ..., \mathbf{Z}_{2k+1}) = \phi(\mathbf{Z}_{k+1})$.

### B.2. Transformers

In this section, we review the basic definitions and notations regarding the transformers architecture. We refer the reader to Phuong & Hutter (2022) for a more extensive overview.

**Attention heads.** We begin by defining *attention heads*. Consider matrices $\mathbf{K}, \mathbf{Q}, \mathbf{V} \in \mathbb{R}^{h \times d}$ and $\mathbf{O} \in \mathbb{R}^{d \times h}$, where $d$ denotes the *embedding dimension* and $h$ denotes the *attention dimension*. Let $\theta := (\mathbf{K}, \mathbf{Q}, \mathbf{V}, \mathbf{O})$ denote the full set of parameters for an attention head. The corresponding attention head $\mathsf{AttHead}_\theta : (\mathbb{R}^d)^L \to \mathbb{R}^d$ takes as input a sequence $\mathbf{h} = (\mathbf{h}_1, ..., \mathbf{h}_L)$ of $L$ embedding vectors corresponding to individual tokens, as well as a *query embedding vector* $\mathbf{h}_{\mathsf{query}} \in \mathbb{R}^d$, and outputs the embedding of the next token, $\mathsf{AttHead}_\theta(\mathbf{h}, \mathbf{h}_{\mathsf{query}})$ by first computing:

$$\mathbf{q} \leftarrow \mathbf{Q} \cdot \mathbf{h}_{\mathsf{query}}$$
$$\mathbf{k}_t \leftarrow \mathbf{K} \cdot \mathbf{h}_t, \quad \mathbf{v}_t \leftarrow \mathbf{V} \cdot \mathbf{h}_t, \quad \alpha_t \leftarrow \sigma(\langle \mathbf{q}, \mathbf{k}_1 \rangle, ..., \langle \mathbf{q}, \mathbf{k}_L \rangle), \quad \forall t \in [L],$$

where typically $\sigma = \mathrm{softmax}$. Then, the attention head outputs $\mathsf{AttHead}_\theta(\mathbf{h}, \mathbf{h}_{\mathsf{query}}) := \mathbf{O} \cdot \sum_{t=1}^{L} \alpha_t \cdot \mathbf{v}_t$.

**Masked self-attention.** We consider transformers which perform masked self-attention, meaning that they aim to predict each token in a sequence by attending to previous tokens in the sequence. Moreover, in doing so, multiple attention heads are combined into an attention layer, of which multiple are combined into a transformer. First, we describe an attention layer: for some $m \in \mathbb{N}$, we consider $m$ attention heads parametrized by $\theta_1, ..., \theta_m$, together with a multilayer perceptron $\mathsf{MLP} : \mathbb{R}^d \to \mathbb{R}^d$, with 2 layers. Letting $\theta = (\theta_1, ..., \theta_h, \mathsf{MLP})$, we let the attention layer $\mathsf{AttLayer}_\theta : (\mathbb{R}^d)^L \to (\mathbb{R}^d)^L$ be the mapping defined as follows: for $i \in [L]$ and $\mathbf{h} = (\mathbf{h}_1, ..., \mathbf{h}_L) \in (\mathbb{R}^d)^L$,

$$\mathsf{AttLayer}_\theta(\mathbf{h})_i := \mathsf{MLP}(\mathbf{h}_i + \mathbf{y}) + \mathbf{h}_i + \mathbf{y}, \quad \mathbf{y} := \sum_{j=1}^{m} \mathsf{AttHead}_{\theta_j}(\mathbf{h}_{1:i}, \mathbf{h}_i).$$

Note that $\mathsf{AttLayer}_\theta(\mathbf{h})_i$ depends only on the embeddings up to index $i$, i.e., $\mathbf{h}_1, ..., \mathbf{h}_i$. Finally, given a collection $\theta = ((\theta^{(1)}, ..., \theta^{(\Lambda)}), \mathbf{W}_{\mathsf{e}})$ denoting parameters for each of $\Lambda \in \mathbb{N}$ layers together with an *embedding matrix* $\mathbf{W}_{\mathsf{e}} \in \mathbb{R}^{d \times |\mathcal{V}|}$, we define $\mathsf{Transformer}_\theta : \mathcal{V}^L \to (\mathbb{R}^d)^L$ by, for $\mathbf{h} = (\mathbf{h}_1, ..., \mathbf{h}_L) \in (\mathbb{R}^d)^L$,

$$\mathsf{Transformer}_\theta(\mathbf{X}) := \mathsf{AttLayer}_{\theta^{(\Lambda)}} \circ \cdots \circ \mathsf{AttLayer}_{\theta^{(1)}}(\mathbf{W}_{\mathsf{e}} \cdot \mathbf{X}),$$

where $\mathbf{X}$ is interpreted as a matrix in $\mathbb{R}^{|\mathcal{V}| \times L}$ whose columns are the one-hot encodings of each of the tokens of $\mathbf{X}$. To produce a sequence of tokens, typically $\mathsf{Transformer}_\theta$ is composed with an *unembedding matrix* $\mathbf{W}_{\mathsf{u}} \in \mathbb{R}^{|\mathcal{V}| \times d}$ to each position, and a distribution for each position may be obtained by applying a softmax.

## B.3. Positional Skip-Wise (PoSE) Training

The *Positional skip-wise (PoSE)* technique (Zhu et al., 2023) (see also (Wu et al., 2024)) aims to ensure that: (a) the position IDs used during training cover all posible position IDs $1, ..., L_{\mathsf{max}}$ that could be observed at test-time (where $L_{\mathsf{max}}$ is the maximum length of a test-time sequence), and (b) the *differences* between different position IDs seen in training sequences is of similar magnitude to that seen in testing sequences. To do so, we fix an integer $c$ denoting a number of *chunks*, and given a sequence of tokens $\mathbf{X} = (\mathbf{X}_1, ..., \mathbf{X}_\ell)$ during training, we partition $\mathbf{X}$ into $c$ contiguous chunks (i.e., subsequences) and assign to each chunk a random contiguous sequence of position IDs so that the first position ID of each chunk is greater than the last position ID of the previous chunk. At test time, one simply uses the true position IDs, namely $(1, 2, ..., L)$, corresponding to a sequence $\mathbf{X}$ of length $L$.

The precise schemes to partition into chunks and assign position IDs that we use are as follows: in all of our experiments we take the number of chunks to be $c = 2$: during training, we split a sequence $\mathbf{X} = (\mathbf{X}_1, ..., \mathbf{X}_\ell)$ into two parts by choosing a uniformly random position to split at. We then choose 2 integers $J_0, J_1 \sim \mathrm{Unif}([\bar{L} - \ell])$, and let the position ID of the first chunk begin at $\min\{J_0, J_1\}$ and the position ID of the second chunk begin at $\max\{J_0, J_1\} + \ell_1$, where $\ell_1$ is the length of the first chunk. This is essentially the same as the scheme used in Zhu et al. (2023) with all $v_i = 0$.

## B.4. Sparse functional attention captures a single attention head

In this section, we prove Proposition B.2, which shows that the $k$-sparse functional attention class in Definition 3.3 is sufficiently general so as to capture the class of single attention heads (as defined in Appendix B.2). First we formally specify the relevant class of attention heads; to ensure that the class maps from sequences of *tokens* (as defined in Definition 3.3) as opposed to sequences of *embedding vectors* (as defined in Appendix B.2), we include an embedding matrix in the definition:

**Definition B.1** (Single attention heads). Fix $d, h \in \mathbb{N}$ and a finite set $\mathcal{V}$ denoting the vocabulary. We define the class $\mathtt{AttHeads}_{d,h,\mathcal{V}}$, which consists of mappings $h : \mathcal{V}^\star \to \mathbb{R}^d$, as follows. It is the set of all mappings from $\mathbf{X} \in \mathcal{V}^L$ to an embedding vector as follows:

$$\mathbf{X} \mapsto \mathsf{AttHead}_\theta(\mathbf{W}_{\mathsf{e}} \cdot \mathbf{X}, \mathbf{h}_{\mathsf{query}}),$$

where $\mathbf{W}$ ranges over all matrices $\mathbf{W} \in \mathbb{R}^{d \times |\mathcal{V}|}$, $\mathbf{h}_{\mathsf{query}}$ ranges over all vectors $\mathbf{h}_{\mathsf{query}} \in \mathbb{R}^d$, and $\theta$ ranges over all tuples $\theta = (\mathbf{K}, \mathbf{Q}, \mathbf{V}, \mathbf{O})$, where $\mathbf{K}, \mathbf{Q}, \mathbf{V} \in \mathbb{R}^{h \times d}$ and $\mathbf{O} \in \mathbb{R}^{d \times h}$. Moreover, we are slightly abusing notation by interpreting $\mathbf{X}$ as the matrix $\mathbf{X} \in \mathbb{R}^{|\mathcal{V}| \times L}$ whose columns are the one-hot vectors of the individual tokens of the sequence $\mathbf{X}$.

**Proposition B.2** (Formal version of Proposition 3.1). *Fix any $d, h \in \mathbb{N}$ and finite set $\mathcal{V}$. Then there are function classes $\mathcal{G}^{\mathsf{key}}$ consisting of functions mapping $\mathbb{N} \times \mathcal{V} \to (\mathbb{R} \cup \{-\infty\})$ and $\mathcal{G}^{\mathsf{val}}$ consisting of functions mapping $\mathcal{V} \to \mathbb{R}^d$ so that $\mathtt{AttHeads}_{d,h,\mathcal{V}}$ is equal to the 1-sparse functional attention class $\mathcal{H}^{\mathsf{attn}}(\mathcal{G}^{\mathsf{key}}, \mathcal{G}^{\mathsf{val}})$.*

*Proof.* We let $\mathcal{G}^{\mathsf{key}}$ be the set of all mappings $g_0 : \mathbb{N} \times \mathcal{V} \to \mathbb{R}$ which only depend on the $\mathcal{V}$-component of the input, and $\mathcal{G}^{\mathsf{val}}$ be the set of all mappings $g_1 : \mathcal{V} \to \mathbb{R}^d$.

Now fix any $\theta = (\mathbf{K}, \mathbf{Q}, \mathbf{V}, \mathbf{O}), \mathbf{h}_{\mathsf{query}}, \mathbf{W}_{\mathsf{e}}$ indexing some element of $\mathtt{AttHeads}_{d,h,\mathcal{V}}$. For any $L \in \mathbb{N}$, given $\mathbf{X} \in \mathbb{R}^{|\mathcal{V}| \times L}$ whose columns represent the one-hot encoding vectors for some length-$L$ sequence, we have

$$\mathsf{AttHead}_\theta(\mathbf{W}_{\mathsf{e}} \cdot \mathbf{X}, \mathbf{h}_{\mathsf{query}}) = \mathbf{O} \cdot \sum_{i=1}^{L} \alpha_i \cdot \mathbf{V} \mathbf{W}_{\mathsf{e}} \mathbf{X}_i$$

$$\text{where} \quad \alpha_i = \mathrm{softmax}(\langle \mathbf{Q} \mathbf{h}_{\mathsf{query}}, \mathbf{K} \mathbf{W}_{\mathsf{e}} \mathbf{X}_1 \rangle, ..., \langle \mathbf{Q} \mathbf{h}_{\mathsf{query}}, \mathbf{K} \mathbf{W}_{\mathsf{e}} \mathbf{X}_L \rangle), \quad i \in [L].$$

Define $g_0 \in \mathcal{G}^{\mathsf{key}}$ by, for $\mathbf{x} \in \mathcal{V}$,

$$g_0(i, \mathbf{x}) = \langle \mathbf{Q} \mathbf{h}_{\mathsf{query}}, \mathbf{K} \mathbf{W}_{\mathsf{e}} \mathbf{x} \rangle,$$

(where we conflate $\mathbf{x} \in \mathcal{V}$ and its one-hot encoding vector in $\mathbb{R}^{|\mathcal{V}|}$), and define $g_1 \in \mathcal{G}^{\mathsf{val}}$ by

$$g_1(\mathbf{x}) := \mathbf{O} \mathbf{V} \mathbf{W}_{\mathsf{e}} \cdot \mathbf{x}.$$

It is now clear that the mapping $\mathbf{X} \to \mathsf{AttHead}_\theta(\mathbf{W}_{\mathsf{e}} \cdot \mathbf{X}, \mathbf{h}_{\mathsf{query}})$ is identical to the 1-sparse functional attention mapping $h_{g_0, g_1}$ defined in Definition 3.3.

Moreover, since we have assumed $h \geq d$, it is evident that any $h_{g_0, g_1}$, for $g_0 \in \mathcal{G}^{\mathsf{key}}$, $g_1 \in \mathcal{G}^{\mathsf{val}}$ may be realized as $\mathbf{X} \mapsto \mathsf{AttHead}_\theta(\mathbf{W}_{\mathsf{e}} \cdot \mathbf{X}, \mathbf{h}_{\mathsf{query}})$ for some $\theta, \mathbf{h}_{\mathsf{query}}, \mathbf{W}_{\mathsf{e}}$ as above. $\square$

## C. Proofs for length generalization

First, we make the following remark on Assumption 4.2.

**Remark C.1** (On Assumption 4.2). To help interpret Assumption 4.2, note first that in order for the distribution ensemble $\mathcal{P}$ to be realizable (Assumption 4.1) by a class $\mathcal{H}^{\mathsf{attn}}(\mathcal{G}^{\mathsf{key}}, \mathcal{G}^{\mathsf{val}})$ satisfying $L_{\mathsf{local}}$-locality (Assumption 3.2), it will typically be the case that $\max_{i \in S^\star} i - \min_{i \in S^\star} i \leq L_{\mathsf{local}}$ with probability at least $1 - \delta$ for $S^\star \in \mathcal{Q}_\ell^{\mathsf{pos}}$, for any $\ell \in \mathbb{N}$. Thus, a natural choice for $\mathcal{Q}_\ell^{\mathsf{pos}}$ is to fix some distribution $\mathcal{Q}_{L_{\mathsf{local}}}^{\mathsf{pos}}$ over sets in $\mathrm{Sets}_k([L_{\mathsf{local}}])$ and let $\mathcal{Q}_\ell^{\mathsf{pos}}$ be the distribution of a random shift of a sample from $\mathcal{Q}_{L_{\mathsf{local}}}^{\mathsf{pos}}$. Formally, $\mathcal{Q}_\ell^{\mathsf{pos}}$ is the distribution of the shift $S + \Delta$, where $S \sim \mathcal{Q}_{L_{\mathsf{local}}}^{\mathsf{pos}}$ and $\Delta \sim \mathrm{Unif}(\{0, 1, ..., \ell - L_{\mathsf{local}}\})$. It is straightforward to see that such a construction ensures that Assumption 4.2 is satisfied with $\eta_\ell = \ell$. More broadly, any value $\eta_\ell \leq \mathrm{poly}(\ell)$ leads to interesting conclusions in the context of our results, so we interpret Assumption 4.2 as being fairly mild.

### C.1. Proof of Theorem 4.3

*Proof of Theorem 4.3.* Write $\mathcal{H}^{\mathsf{attn}} := \mathcal{H}^{\mathsf{attn}}(\mathcal{G}^{\mathsf{key}}, \mathcal{G}^{\mathsf{val}})$. By Assumption 4.1, we have $\mathbb{E}_{(\mathbf{X}, \mathbf{Y}) \sim \mathcal{P}_\ell}[\mathcal{L}(h^\star(\mathbf{X}), \mathbf{Y})] \leq \delta$ for some $h^\star \in \mathcal{H}^{\mathsf{attn}}$. Let $\hat{h} \in \mathcal{H}^{\mathsf{attn}}$ be chosen according to (1), so that we can write $\hat{h} = h_{\hat{g}_0, \hat{g}_1}$ for some $\hat{g}_0 \in \mathcal{G}^{\mathsf{key}}, \hat{g}_1 \in \mathcal{G}^{\mathsf{val}}$. Thus, for each $\ell \in [L/2, L]$, we have

$$L\delta \geq \mathbb{E}_{(\mathbf{X}, \mathbf{Y}) \sim \mathcal{P}_\ell}[\mathcal{L}(\hat{h}(\mathbf{X}), \mathbf{Y})] \tag{4}$$

$$= \mathbb{E}_{(\mathbf{X}, \mathbf{Y}) \sim \mathcal{P}_\ell}\left[\left\|\mathbf{Y} - \frac{\sum_{S \in \mathrm{Sets}_k([\ell])} \exp(\hat{g}_0(S, \mathbf{X}_S)) \cdot \hat{g}_1(\mathbf{X}_S)}{\sum_{S' \in \mathrm{Sets}_k([\ell])} \exp(\hat{g}_0(S', \mathbf{X}_{S'}))}\right\|\right]$$

$$= \mathbb{E}_{(\mathbf{X}, S^\star) \sim \mathcal{P}_\ell}\left[\left\|g^\star(\mathbf{X}_{S^\star}) - \frac{\sum_{S \in \mathrm{Sets}_k([\ell])} \exp(\hat{g}_0(S, \mathbf{X}_S)) \cdot \hat{g}_1(\mathbf{X}_S)}{\sum_{S' \in \mathrm{Sets}_k([\ell])} \exp(\hat{g}_0(S', \mathbf{X}_{S'}))}\right\|\right], \tag{5}$$

where $g^\star : \mathcal{V}^k \to \mathcal{Y}$ is the function guaranteed by Definition 3.2. By Assumption 4.2, for any $S^\star \in \mathrm{Sets}_k([L])$, we have $\frac{\mathcal{Q}_{L-2L_{\mathrm{local}}}(S^\star)}{\mathcal{Q}_L(S^\star)} \leq \eta_L$ and $\frac{\mathcal{Q}_{L-L_{\mathrm{local}}}(S^\star)}{\mathcal{Q}_L(S^\star)} \leq \eta_L$. (Recall that if, e.g., $S^\star \notin \mathrm{Sets}_k([L-L_{\mathrm{local}}])$, then we use the convention that $\mathcal{Q}_{L-L_{\mathrm{local}}}(S^\star) = 0$.) At various points later in the proof, we will need to use (5) for $\ell = L - L_{\mathrm{local}}$ or $\ell = L - 2L_{\mathrm{local}}$; in order for this to be valid, we need $L - 2L_{\mathrm{local}} \geq L/2$, i.e., $L \geq 4L_{\mathrm{local}}$, which is ensured by the hypotheses of Theorem 4.3.

Thus, by the definition of the $k$-sparse planted structure (Definition 3.2; in particular, we are using the fact that $\mathbf{X}_i \sim \mu$ for all $i \notin S^\star$), we have that, for any $\mathbf{X} \in \mathcal{V}^L, S^\star \in \mathrm{Sets}_k([L])$,

$$\frac{\mathcal{P}_{L-2L_{\mathrm{local}}}(\mathbf{X}_{1:L-2L_{\mathrm{local}}}, S^\star) \cdot \prod_{i=L-2L_{\mathrm{local}}+1}^{L} \mu(\mathbf{X}_i)}{\mathcal{P}_L(\mathbf{X}, S^\star)} \leq \eta_L, \qquad \frac{\mathcal{P}_{L-2L_{\mathrm{local}}}(\mathbf{X}_{1:L-2L_{\mathrm{local}}}, S^\star) \cdot \prod_{i=L-2L_{\mathrm{local}}+1}^{L-L_{\mathrm{local}}} \mu(\mathbf{X}_i)}{\mathcal{P}_{L-L_{\mathrm{local}}}(\mathbf{X}, S^\star)} \leq \eta_L. \quad (6)$$

It follows that

$$\eta_L \cdot (L\delta) \geq \mathop{\mathbb{E}}_{\substack{(\mathbf{X}_{1:L-2L_{\mathrm{local}}}, S^\star) \sim \mathcal{P}_{L-2L_{\mathrm{local}}} \\ \mathbf{X}_{L-2L_{\mathrm{local}}+1:L} \sim \mu^{\otimes 2L_{\mathrm{local}}}}} \left[ \left\| g^\star(\mathbf{X}_{S^\star}) - \frac{\sum_{S \in \mathrm{Sets}_k([L])} \exp(\hat{g}_0(S, \mathbf{X}_S)) \cdot \hat{g}_1(\mathbf{X}_S)}{\sum_{S' \in \mathrm{Sets}_k([L])} \exp(\hat{g}_0(S', \mathbf{X}_{S'}))} \right\| \right] \quad (7)$$

$$\eta_L \cdot (L\delta) \geq \mathop{\mathbb{E}}_{\substack{(\mathbf{X}_{1:L-2L_{\mathrm{local}}}, S^\star) \sim \mathcal{P}_{L-2L_{\mathrm{local}}} \\ \mathbf{X}_{L-2L_{\mathrm{local}}+1:L-L_{\mathrm{local}}} \sim \mu^{\otimes L_{\mathrm{local}}}}} \left[ \left\| g^\star(\mathbf{X}_{S^\star}) - \frac{\sum_{S \in \mathrm{Sets}_k([L-L_{\mathrm{local}}])} \exp(\hat{g}_0(S, \mathbf{X}_S)) \cdot \hat{g}_1(\mathbf{X}_S)}{\sum_{S' \in \mathrm{Sets}_k([L-L_{\mathrm{local}}])} \exp(\hat{g}_0(S', \mathbf{X}_{S'}))} \right\| \right]. \quad (8)$$

The triangle inequality then gives

$$2\eta_L \cdot (L\delta) \geq \mathop{\mathbb{E}}_{\substack{(\mathbf{X}_{1:L-2L_{\mathrm{local}}}, S^\star) \sim \mathcal{P}_{L-2L_{\mathrm{local}}} \\ \mathbf{X}_{L-2L_{\mathrm{local}}+1:L} \sim \mu^{\otimes 2L_{\mathrm{local}}}}} \left[ \left\| \frac{\sum_{S \in \mathrm{Sets}_k([L])} \exp(\hat{g}_0(S, \mathbf{X}_S)) \cdot \hat{g}_1(\mathbf{X}_S)}{\sum_{S' \in \mathrm{Sets}_k([L])} \exp(\hat{g}_0(S', \mathbf{X}_{S'}))} - \frac{\sum_{S \in \mathrm{Sets}_k([L-L_{\mathrm{local}}])} \exp(\hat{g}_0(S, \mathbf{X}_S)) \cdot \hat{g}_1(\mathbf{X}_S)}{\sum_{S' \in \mathrm{Sets}_k([L-L_{\mathrm{local}}])} \exp(\hat{g}_0(S', \mathbf{X}_{S'}))} \right\| \right].$$
$$(9)$$

We define the following random variables, where $(\mathbf{X}_{1:L-2L_{\mathrm{local}}}, S^\star) \sim \mathcal{P}_{L-2L_{\mathrm{local}}}$, $\mathbf{X}_j \sim \mu$ for $j > L - 2L_{\mathrm{local}}$, and $i \geq 1$:

$$A_0 := \sum_{S \in \mathrm{Sets}_k([L-2L_{\mathrm{local}}])} \exp(\hat{g}_0(S, \mathbf{X}_S)) \cdot \hat{g}_1(\mathbf{X}_S), \qquad B_0 := \sum_{S' \in \mathrm{Sets}_k([L-2L_{\mathrm{local}}])} \exp(\hat{g}_0(S', \mathbf{X}_{S'}))$$

$$A_i = \sum_{S \in \mathrm{Sets}_k([L-2L_{\mathrm{local}}+iL_{\mathrm{local}}]) \setminus \mathrm{Sets}_k([L-2L_{\mathrm{local}}+(i-1)L_{\mathrm{local}}])} \exp(\hat{g}_0(S, \mathbf{X}_S)) \cdot \hat{g}_1(\mathbf{X}_S),$$

$$B_i := \sum_{S \in \mathrm{Sets}_k([L-2L_{\mathrm{local}}+iL_{\mathrm{local}}]) \setminus \mathrm{Sets}_k([L-2L_{\mathrm{local}}+(i-1)L_{\mathrm{local}}])} \exp(\hat{g}_0(S, \mathbf{X}_S)).$$

Note that $A_0, A_1, \ldots \in \mathbb{R}^d$ are vectors, and $B_0, B_1, \ldots \in \mathbb{R}$ are scalars. We have that

$$\mathbb{E}\left[ \left\| \frac{A_0}{B_0} - \frac{A_0 + A_1}{B_0 + B_1} \right\| \right] \leq 2\eta_L \cdot (L\delta), \qquad \mathbb{E}\left[ \left\| \frac{A_0 + A_1 + A_2}{B_0 + B_1 + B_2} - \frac{A_0 + A_1}{B_0 + B_1} \right\| \right] \leq 2\eta_L \cdot (L\delta),$$

where the first inequality uses (5) with $\ell = L - 2L_{\mathrm{local}}$, (8) and the triangle inequality, and the second inequality uses (9). Let $Z_0$ denote the collection of random variables $\mathbf{X}_{1:L-2L_{\mathrm{local}}}$, and for $i \geq 1$, let $Z_i$ denote the collection of random variables $\mathbf{X}_{L-2L_{\mathrm{local}}+(i-1)L_{\mathrm{local}}:L-2L_{\mathrm{local}}+iL_{\mathrm{local}}}$. Then $Z_0, Z_1, \ldots$ are independent, and $Z_1, Z_2, \ldots$ are identically distributed (since we have $(\mathbf{X}_{1:L-2L_{\mathrm{local}}}, S^\star) \sim \mathcal{P}_{L-2L_{\mathrm{local}}}$). Since $\hat{g}_0(S, \mathbf{X}_S) = -\infty$ for any $S$ for which $\max(S) - \min(S) > L_{\mathrm{local}}$ (Item 1 of Assumption 3.2), $A_i, B_i$ can each be written as functions of $Z_i, Z_{i-1}$ for each $i \geq 0$. Moreover, by Item 2 of Assumption 3.2, this function does not depend on $i$ for $i \geq 1$. By Lemma C.1, it follows that, for any $t \geq -2$,

$$\mathop{\mathbb{E}}_{\substack{(\mathbf{X}_{1:L-2L_{\mathrm{local}}}, S^\star) \sim \mathcal{P}_{L-2L_{\mathrm{local}}} \\ \mathbf{X}_j \sim \mu \ \forall j > L-2L_{\mathrm{local}}}} \left[ \left\| \frac{\sum_{S \in \mathrm{Sets}_k([L-2L_{\mathrm{local}}])} \exp(\hat{g}_0(S, \mathbf{X}_S)) \cdot \hat{g}_1(\mathbf{X}_S)}{\sum_{S' \in \mathrm{Sets}_k([L-2L_{\mathrm{local}}])} \exp(\hat{g}_0(S', \mathbf{X}_{S'}))} - \frac{\sum_{S \in \mathrm{Sets}_k([L+tL_{\mathrm{local}}])} \exp(\hat{g}_0(S, \mathbf{X}_S)) \cdot \hat{g}_1(\mathbf{X}_S)}{\sum_{S' \in \mathrm{Sets}_k([L+tL_{\mathrm{local}}])} \exp(\hat{g}_0(S', \mathbf{X}_{S'}))} \right\| \right]$$
$$\leq 6\eta_L(L\delta)(t+2).$$

A symmetric argument establishes that for any $t \geq 0$,

$$\mathop{\mathbb{E}}_{\substack{(\mathbf{X}_{1:L-2L_{\mathrm{local}}}, S^\star) \sim \mathcal{P}_{L-2L_{\mathrm{local}}} \\ \mathbf{X}_j \sim \mu \ \forall j \leq 0}} \left[ \left\| \frac{\sum_{S \in \mathrm{Sets}_k([L-2L_{\mathrm{local}}])} \exp(\hat{g}_0(S, \mathbf{X}_S)) \cdot \hat{g}_1(\mathbf{X}_S)}{\sum_{S' \in \mathrm{Sets}_k([L-2L_{\mathrm{local}}])} \exp(\hat{g}_0(S', \mathbf{X}_{S'}))} - \frac{\sum_{S \in \mathrm{Sets}_k([-tL_{\mathrm{local}}+1, L-2L_{\mathrm{local}}])} \exp(\hat{g}_0(S, \mathbf{X}_S)) \cdot \hat{g}_1(\mathbf{X}_S)}{\sum_{S' \in \mathrm{Sets}_k([-tL_{\mathrm{local}}+1, L-2L_{\mathrm{local}}])} \exp(\hat{g}_0(S', \mathbf{X}_{S'}))} \right\| \right]$$
$$\leq 6\eta_L(L\delta) \cdot t.$$

Combining the two displays above with Lemma C.2 (and again using Item 1 of Assumption 3.2), we see that for any $t_0 \geq 0, t_1 \geq -2$,

$$
\mathbb{E}_{\substack{(\mathbf{X}_{1:L-2L_{\mathsf{local}}}, S^\star) \sim \mathcal{P}_{L-2L_{\mathsf{local}}} \\ \mathbf{X}_j \sim \mu \; \forall j \leq 0, j > L - 2L_{\mathsf{local}}}} \left[ \left\| \frac{\sum_{S \in \mathrm{Sets}_k([L-2L_{\mathsf{local}}])} \exp(\hat{g}_0(S, \mathbf{X}_S)) \cdot \hat{g}_1(\mathbf{X}_S)}{\sum_{S' \in \mathrm{Sets}_k([L-2L_{\mathsf{local}}])} \exp(\hat{g}_0(S', \mathbf{X}_{S'}))} - \frac{\sum_{S \in \mathrm{Sets}_k([-t_0 L_{\mathsf{local}}+1, L+t_1 L_{\mathsf{local}}])} \exp(\hat{g}_0(S, \mathbf{X}_S)) \cdot \hat{g}_1(\mathbf{X}_S)}{\sum_{S' \in \mathrm{Sets}_k([-t_0 L_{\mathsf{local}}+1, L+t_1 L_{\mathsf{local}}])} \exp(\hat{g}_0(S', \mathbf{X}_{S'}))} \right\| \right]
$$
$$
\leq 6 \eta_L (L\delta)(t_0 + t_1 + 2). \tag{10}
$$

Let $\mathcal{P}_{L-2L_{\mathsf{local}}, t_0, t_1}$ denote the distribution of $\tilde{\mathbf{X}}_{1:L+L_{\mathsf{local}}(t_0+t_1)} \in \mathcal{V}^{L+L_{\mathsf{local}}(t_0+t_1)}$ and $\tilde{S}^\star \in \mathrm{Sets}_k([L+L_{\mathsf{local}}(t_0+t_1)])$ defined as follows: first, draw $(\mathbf{X}_{1:L-L_{\mathsf{local}}}, S^\star) \sim \mathcal{P}_{L-2L_{\mathsf{local}}}$, as well as $\mathbf{X}_j \sim \mu$ for all $j \leq 0$, $j > L-2L_{\mathsf{local}}$, and then set $\tilde{\mathbf{X}}_j = \mathbf{X}_{j-t_0 L_{\mathsf{local}}}$ for $j \in [L+L_{\mathsf{local}}(t_0+t_1)]$ and $\tilde{S}^\star = S^\star + t_0 L_{\mathsf{local}}$. Then (10) together with (5) (with $\ell = L - 2L_{\mathsf{local}}$) gives that, for all $t_0 \geq 0, t_1 \geq -2$,

$$
\mathbb{E}_{(\mathbf{X}_{1:L+L_{\mathsf{local}}(t_0+t_1)}, S^\star) \sim \mathcal{P}_{L-2L_{\mathsf{local}}, t_0, t_1}} \left[ \left\| g^\star(\mathbf{X}_{S^\star}) - \frac{\sum_{S \in \mathrm{Sets}_k([L+L_{\mathsf{local}}(t_0+t_1)])} \exp(\hat{g}_0(S, \mathbf{X}_S)) \cdot \hat{g}_1(\mathbf{X}_S)}{\sum_{S' \in \mathrm{Sets}_k([L+L_{\mathsf{local}}(t_0+t_1)])} \exp(\hat{g}_0(S', \mathbf{X}_{S'}))} \right\| \right]
$$
$$
\leq O(\eta_L \cdot (L\delta) \cdot (t_0 + t_1 + 2)). \tag{11}
$$

By Definition 3.2 and Assumption 4.2, for any $\mathbf{X}_{1:\bar{L}} \in \mathcal{V}^{\bar{L}}$ and $S^\star \in \mathrm{Sets}_k([\bar{L}])$ with $\max\{S^\star\} - \min\{S^\star\} \leq L_{\mathsf{local}}$, there are some $t_0 \geq 0, t_1 \geq -2$ satisfying $t_0 + t_1 + 2 = (\bar{L} - L)/L_{\mathsf{local}}$[11] so that

$$
\frac{\mathcal{P}_{\bar{L}}(\mathbf{X}_{1:\bar{L}}, S^\star)}{\mathcal{P}_{L-2L_{\mathsf{local}}, t_0, t_1}(\mathbf{X}_{1:\bar{L}}, S^\star)} \leq \frac{\mathcal{Q}_{\bar{L}}(S^\star)}{\mathcal{Q}_{L-2L_{\mathsf{local}}}(S^\star - t_0 \cdot L_{\mathsf{local}})} \leq \eta_{\bar{L}}. \tag{12}
$$

Noting that we have $\bar{L} = L + L_{\mathsf{local}}(t_0 + t_1)$, using (12) and summing (11) over all possible values of $t_0, t_1$ for which $t_0 + t_1 + 2 = (\bar{L} - L)/L_{\mathsf{local}}$ gives that

$$
\mathbb{E}_{(\mathbf{X}_{1:\bar{L}}, S^\star) \sim \mathcal{P}_{\bar{L}}} \left[ \left\| g^\star(\mathbf{X}_{S^\star}) - \frac{\sum_{S \in \mathrm{Sets}_k([\bar{L}])} \exp(\hat{g}_0(S, \mathbf{X}_S)) \cdot \hat{g}_1(\mathbf{X}_S)}{\sum_{S' \in \mathrm{Sets}_k([\bar{L}])} \exp(\hat{g}_0(S', \mathbf{X}_{S'}))} \right\| \right]
$$
$$
\leq O(\eta_L \eta_{\bar{L}} \cdot (L\delta) \cdot (t_0 + t_1)^2).
$$

Using that $g^\star(\mathbf{X}_{S^\star}) = \mathbf{Y}$ under $(\mathbf{X}_{1:\bar{L}}, S^\star, \mathbf{Y}) \sim \mathcal{P}_{\bar{L}}$, we conclude that

$$
\mathbb{E}_{(\mathbf{X}, \mathbf{Y}) \sim \mathcal{P}_{\bar{L}}} \left[ \mathcal{L}(\hat{h}(\mathbf{X}), \mathbf{Y}) \right] \leq O(\eta_L \eta_{\bar{L}} \cdot L \cdot (\bar{L}/L_{\mathsf{local}})^2 \cdot \delta).
$$

$\square$

**Lemma C.1.** *Suppose that $Z_0, Z_1, Z_2, \ldots$ are independent random variables with sample space $\Omega$, and that $Z_1, Z_2, \ldots$ are identically distributed. Furthermore suppose $f, f_0 : \Omega \to \mathbb{R}^d$ and $g, g_0 : \Omega \to \mathbb{R}$ are measurable functions, and let $A_i = f(Z_i, Z_{i-1}), B_i = g(Z_i, Z_{i-1})$ for each $i \geq 1$, and $A_0 = f_0(Z_1, Z_0), B_0 = g_0(Z_1, Z_0)$. Suppose $\epsilon > 0$ satisfies*

$$
\mathbb{E}\left[ \left\| \frac{A_0}{B_0} - \frac{A_0 + A_1}{B_0 + B_1} \right\| \right] \leq \epsilon, \quad \mathbb{E}\left[ \left\| \frac{A_0 + A_1}{B_0 + B_1} - \frac{A_0 + A_1 + A_2}{B_0 + B_1 + B_2} \right\| \right] \leq \epsilon. \tag{13}
$$

*Then for any $i \geq 1$, we have*

$$
\mathbb{E}\left[ \left\| \frac{A_0}{B_0} - \frac{A_0 + A_1 + \cdots + A_i}{B_0 + B_1 + \cdots + B_i} \right\| \right] \leq 3i\epsilon.
$$

*Proof.* We use a hybrid argument. Fix any $i \geq 2$, and define $A_1' = f(Z_1, Z_i), B_1' = g(Z_1, Z_i)$. We have

$$
\mathbb{E}\left[ \left\| \frac{A_0}{B_0} - \frac{A_0 + A_1 + A_i}{B_0 + B_1 + B_i} \right\| \right]
$$
$$
\leq \mathbb{E}\left[ \left\| \frac{A_0}{B_0} - \frac{A_0 + A_1'}{B_0 + B_1'} \right\| \right] + \mathbb{E}\left[ \left\| \frac{A_0 + A_1'}{B_0 + B_1'} - \frac{A_0 + A_1 + A_i}{B_0 + B_1 + B_i} \right\| \right] \leq 2\epsilon,
$$

---

[11] In particular, any choice of $t_0$ satisfying $S^\star - t_0 \cdot L_{\mathsf{local}} \in \mathrm{Sets}_k([L-2L_{\mathsf{local}}])$ suffices.

where the final inequality follows from (13) together with the fact that $Z_1, Z_2, Z_i$ are iid. By the triangle inequality, it follows that

$$\mathbb{E}\left[\left\|\frac{A_0+A_1}{B_0+B_1}-\frac{A_0+A_1+A_i}{B_0+B_1+B_i}\right\|\right]\leq 3\epsilon.$$

Fix any $j\geq 2$. Summing over $2\leq i\leq j$ and using Lemma C.2, we see that

$$\mathbb{E}\left[\left\|\frac{A_0+A_1}{B_0+B_1}-\frac{A_0+A_1+\cdots+A_j}{B_0+B_1+\cdots+B_j}\right\|\right]\leq 3\epsilon\cdot(j-1).$$

A final application of the triangle inequality yields the conclusion of the lemma. $\qquad\square$

**Lemma C.2.** *Let $L\in\mathbb{N}$ and $a,q_i\in\mathbb{R}^d$ be vectors for $1\leq i\leq L$ and $b,p_i\in\mathbb{R}_{>0}$ be positive real numbers for $1\leq i\leq L$. Suppose that for each $i\in[L]$, $\left\|\frac{a}{b}-\frac{q_i+a}{p_i+b}\right\|=\epsilon_i$. Then*

$$\left\|\frac{a}{b}-\frac{\sum_{i=1}^L q_i+a}{\sum_{i=1}^L p_i+b}\right\|\leq\sum_{i=1}^L\epsilon_i.$$

*Proof.* For each $i$, we have:

$$\left\|\frac{p_i a-b q_i}{b(p_i+b)}\right\|=\left\|\frac{a}{b}-\frac{q_i+a}{p_i+b}\right\|\leq\epsilon_i.$$

Multiplying both sides by $b(p_i+b)$ (which is positive), we see that

$$\|p_i a-q_i b\|\leq\epsilon_i b(p_i+b).$$

Summing both sides over $i=1$ to $L$:

$$\left\|a\sum_{i=1}^L p_i-b\sum_{i=1}^L q_i\right\|=\left\|\sum_{i=1}^L p_i a-q_i b\right\|\leq b\sum_{i=1}^L\epsilon_i(p_i+b).$$

Let $\epsilon=\sum_{i=1}^L\epsilon_i$, $S=\sum_{i=1}^L p_i$ and $V=\sum_{i=1}^L q_i$, so $\|aS-bV\|\leq\epsilon b(S+b)$. We need to bound $\left\|\frac{a}{b}-\frac{V+a}{S+b}\right\|$. To do so, we compute

$$\left\|\frac{a}{b}-\frac{V+a}{S+b}\right\|=\left\|\frac{aS-bV}{b(S+b)}\right\|\leq\frac{\epsilon b(S+b)}{b(S+b)}=\epsilon,$$

which gives the desired bound. $\qquad\square$

### C.2. Improving the length generalization error

Below we discuss some ways that the error bound of $O(\eta_L\eta_{\bar{L}}\cdot L\cdot(\bar{L}/L_{\mathsf{local}})^2\cdot\delta)$ in Theorem 4.3 can be improved:

- First, the factor of $L$ can be removed if instead of the training objective (1) we choose

$$\hat{h}:=\underset{h\in\mathcal{H}}{\arg\min}\max_{\ell\in[L/2,L]}\underset{(\mathbf{X},\mathbf{Y})\sim\mathcal{P}_\ell}{\mathbb{E}}[\mathcal{L}(h(\mathbf{X}),\mathbf{Y})].$$

  (In particular, we take the worst length $\ell\in[L/2,L]$, as opposed to an average length.) This holds because the factor of $L$ is picked up in the first step of the proof (4) where we bound the in-distribution loss for each $\ell\in[L/2,L]$. We opt to use the formulation in (1) in the formal statement of Theorem 4.3 since it more closely correponds to what is done in practice.

- When applying Theorem 4.3 it is without loss to take $L_{\mathsf{local}}=L/4$, since making $L_{\mathsf{local}}$ larger only make Assumption 4.2 easier to satisfy. Thus, the term $(\bar{L}/L_{\mathsf{local}})^2$ will be bounded as $O((\bar{L}/L)^2)$.

- If we take $\mathcal{Q}_\ell^{\mathsf{pos}}$ to be as in Remark C.1, then we can use the shift-invariance of the distributions $\mathcal{Q}_\ell^{\mathsf{pos}}$ to remove the term $\eta_L\cdot\eta_{\bar{L}}$ from the error bound as follows. The term $\eta_L$ in (6) may be replaced with $O(1)$ since $L-2L_{\mathsf{local}}\geq L/2$. Moreover, the instance of $\eta_{\bar{L}}$ in (12) can be replaced with $O(1)$ since $\bar{L}\geq L-2L_{\mathsf{local}}$.

If we incorporate all of the above optimizations, then we obtain that (under the assumption on $\mathcal{Q}_\ell^{\mathsf{pos}}$ introduced in Remark C.1), the error bound in Theorem 4.3 can be replaced with $O((\bar{L}/L)^2\cdot\delta)$.

## C.3. On necessity of the assumptions

In the below proposition, we establish that each of the assumptions of Theorem 4.3 is necessary in that none of them can be individually removed.

**Proposition C.3.** *There is a constant $c > 0$ so that the following holds. For any of the assumptions: (a) Item 1 of Assumption 3.2, (b) Item 2 of Assumption 3.2, (c) Assumption 4.1 with $\delta = 0$, (d) Assumption 4.2 with $\eta_\ell = O(\ell)$, and any integer $L \in \mathbb{N}$, there is a 2-sparse functional attention class $\mathcal{H}^{\mathsf{attn}} = \mathcal{H}^{\mathsf{attn}}(\mathcal{G}^{\mathsf{key}}, \mathcal{G}^{\mathsf{val}})$ and a 2-sparse planted correlations distribution ensemble $\mathcal{P}$ which satisfy each of the other 3 assumptions with $L_{\mathsf{local}} = 2$ but which does not have $(L, \bar{L}, c)$-length generalization for any $\bar{L} \geq 2L$.*

*Proof.* We show in turn that each of the assumptions cannot be removed. For all of the examples, we set $k = 2$, and $\mathcal{V} = \{1, 2\}$. We let $\mathcal{Y} = [0,1]$ be the unit interval, and let $\mathcal{L}(\mathbf{Y}, \mathbf{Y}') := |\mathbf{Y} - \mathbf{Y}'|$.

**Removing locality.** We first show that $L_{\mathsf{local}}$-locality (Item 1 of Assumption 3.2) cannot be dropped. We consider the sparse function attention class $\mathcal{H}^{\mathsf{attn}} = \mathcal{H}^{\mathsf{attn}}(\mathcal{G}^{\mathsf{key}}, \mathcal{G}^{\mathsf{val}})$ where $\mathcal{G}^{\mathsf{key}}, \mathcal{G}^{\mathsf{val}}$ are defined as follows:

- $\mathcal{G}^{\mathsf{key}} = \{g_{00}, g_{01}\}$, defined as follows:

$$g_{00}(S, \mathbf{x}) = \begin{cases} 0 & : \mathbf{x} = (2,2) \text{ or } \max\{S\} - \min\{S\} \geq L \\ -\infty & : \text{otherwise} \end{cases}$$

$$g_{01}(S, \mathbf{x}) = \begin{cases} 0 & : \mathbf{x} = (2,2) \\ -\infty & : \text{otherwise.} \end{cases}$$

- $\mathcal{G}^{\mathsf{val}}$ consists of the single function $g_1 : \mathcal{V}^2 \to \mathcal{Y}$ defined as $g_1(2,2) = 1$ and $g_1(\mathbf{x}_1, \mathbf{x}_2) = 0$ for all $(\mathbf{x}_1, \mathbf{x}_2) \neq (2,2)$.

Fix any integer $L$. We define a distribution ensemble $\mathcal{P}$ with $k$-sparse planted correlations, as specified by distributions $\mu \in \Delta(\mathcal{V}), \mathcal{Q}_\ell^{\mathsf{pos}} \in \Delta(\mathsf{Sets}_k([\ell]))$ for each $\ell \in \mathbb{N}$, and $\mathcal{Q}^{\mathsf{voc}} \in \Delta(\mathcal{V}^k)$, as well as a function $g^\star : \mathcal{V}^k \to \mathcal{Y}$, as follows:

- $\mu$ is the point mass on $\{1\}$.

- $\mathcal{Q}^{\mathsf{voc}}$ is the point mass on $\{(2,2)\}$.

- $\mathcal{Q}_\ell^{\mathsf{pos}}$ is uniform over the sets $\{1, 2\}, \{2, 3\}, \ldots, \{\ell - 1, \ell\}$ if $\ell \leq L$; and is uniform over the sets $\{1,2\}, \ldots, \{\ell-1, \ell\}, \{1, L+1\}, \{2, L+2\}, \ldots, \{\ell - L, \ell\}$ if $\ell > L$.

- We set $g^\star = g_1 \in \mathcal{G}^{\mathsf{val}}$ as defined above.

It is immediate from the above definitions that $\mathcal{H}^{\mathsf{attn}}$ is relative (i.e., Item 2 of Assumption 3.2). Next, realizability (Assumption 4.1) is clearly satisfied as the function $h_{g_{00}, g_1}$ achieves 0 loss. Finally, coverage (Assumption 4.2) is readily verified by taking $\eta_\ell = 2\ell$ and $L_{\mathsf{local}} = 2$. (In particular, for $S^\star = \{a, a+L\} \in \mathsf{Sets}_k([\ell])$ and $\ell' \leq L$, we have that $S^\star - \Delta \notin \mathsf{Sets}_k([\ell'])$ for all choices of $\Delta$ and so we do not need to satisfy $\frac{\mathcal{Q}_\ell^{\mathsf{pos}}(S^\star)}{\mathcal{Q}_{\ell'}^{\mathsf{pos}}(S^\star - \Delta)} \leq \eta_\ell$ (such an inequality does not even make sense).

Finally, note that the risk minimization procedure which returns $\hat{h} = h_{g_{00}, g_1}$ satisfies (1) with the chosen value of $L$ and error $\delta = 0$. However, it is readily checked that $\hat{h}$ experiences error $\Omega(1)$ on the distribution $\mathcal{P}_\ell$ for any $\ell \geq 2L$.

**Removing the relative assumption.** Next we show that Item 2 of Assumption 3.2 cannot be dropped. We consider the sparse attention class defined by the following $\mathcal{G}^{\mathsf{key}}, \mathcal{G}^{\mathsf{val}}$:

- $\mathcal{G}^{\mathsf{key}} = \{g_{00}, g_{01}\}$, defined as follows:

$$g_{00}(S, \mathbf{x}) = \begin{cases} 0 & : \mathbf{x} = (2,2) \text{ and } \max\{S\} - \min\{S\} = 1 \\ -\infty & : \text{otherwise} \end{cases}$$

$$g_{01}(S, \mathbf{x}) = \begin{cases} 0 & : \mathbf{x} = (2,2) \text{ and } \max\{S\} - \min\{S\} = 1 \\ 0 & : \max\{S\} - \min\{S\} = 2, \max\{S\} > L \\ -\infty & : \text{otherwise.} \end{cases}$$

- $\mathcal{G}^{\mathsf{val}}$ consists of the single function $g_1 : \mathcal{V}^2 \to \mathcal{Y}$ defined as $g_1(2,2) = 1$ and $g_1(\mathbf{x}_1,\mathbf{x}_2) = 0$ for all $(\mathbf{x}_1,\mathbf{x}_2) \neq (2,2)$.

Fix an integer $L$. We define an ensemble $\mathcal{P}$ specified by $\mu, \mathcal{Q}_\ell^{\mathsf{pos}}, \mathcal{Q}^{\mathsf{voc}}, g^\star$ as follows:

- $\mu$ is the point mass on $\{1\}$.

- $\mathcal{Q}^{\mathsf{voc}}$ is the point mass on $\{(2,2)\}$.

- $\mathcal{Q}_\ell^{\mathsf{pos}}$ is uniform over the sets $\{1,2\},\{2,3\},...,\{\ell-1,\ell\}$.

- $g^\star = g_1 \in \mathcal{G}^{\mathsf{val}}$.

It is immediate from the above definitions that $\mathcal{H}^{\mathsf{attn}} = \mathcal{H}^{\mathsf{attn}}(\mathcal{G}^{\mathsf{key}},\mathcal{G}^{\mathsf{val}})$ is $L_{\mathsf{local}}$-local (Item 1 of Assumption 3.2) with $L_{\mathsf{local}} = 2$. Realizability (Assumption 4.1) is satisfied with $h^\star = h_{g_{00},g_1}$. Coverage (Assumption 4.2) is satisfied with $L_{\mathsf{local}} = 1$ and $\eta_\ell = \ell$.

The risk minimization procedure which returns $\hat{h} = h_{g_{01},g_1}$ satisfies (1) with the chosen value of $L$ and error $\delta = 0$, but $\hat{h}$ experiences error $\Omega(1)$ on the distribution $\mathcal{P}_\ell$ for any $\ell \geq L+1$.

**Removing realizability.** Next, we show that Assumption 4.1 cannot be dropped. We consider the sparse attention class defined by the following $\mathcal{G}^{\mathsf{key}},\mathcal{G}^{\mathsf{val}}$:

- $\mathcal{G}^{\mathsf{key}} = \{g_{00},g_{01}\}$, defined as follows:

$$g_{00}(S,\mathbf{x}) = \begin{cases} 0 & : \max\{S\}-\min\{S\} = 1 \text{ and } \mathbf{x} = (2,2) \\ -\infty & : \text{otherwise} \end{cases}$$

$$g_{01}(S,\mathbf{x}) = \begin{cases} 0 & : \max\{S\}-\min\{S\} = 2 \text{ and } \mathbf{x} = (2,2) \\ -\infty & : \text{otherwise.} \end{cases}$$

Fix an integer $L$. We define an ensemble $\mathcal{P}$ specified by the following $\mu, \mathcal{Q}_\ell^{\mathsf{pos}}, \mathcal{Q}^{\mathsf{voc}}, g^\star$:

- $\mu$ is the point mass on $\{1\}$.

- $\mathcal{Q}^{\mathsf{voc}}$ is the point mass on $\{(2,2)\}$.

- $\mathcal{Q}_\ell^{\mathsf{pos}}$ is uniform over the sets $\{1,2\},\{2,3\},...,\{\ell-1,\ell\}$ if $\ell \leq L$. For $\ell > L$, we define:

$$\mathcal{Q}_\ell^{\mathsf{pos}}(\{1,2\}) = \cdots = \mathcal{Q}_\ell^{\mathsf{pos}}(\{\ell-1,\ell\}) = \frac{1}{10(\ell-1)}$$

$$\mathcal{Q}_\ell^{\mathsf{pos}}(\{1,3\}) = \mathcal{Q}_\ell^{\mathsf{pos}}(\{2,4\}) = \cdots = \mathcal{Q}_\ell^{\mathsf{pos}}(\{\ell-2,\ell\}) = \frac{9}{10(\ell-2)}. \tag{14}$$

- $g^\star = g_1 \in \mathcal{G}^{\mathsf{val}}$.

It is immediate from the above definitions that $\mathcal{H}^{\mathsf{attn}} = \mathcal{H}^{\mathsf{attn}}(\mathcal{G}^{\mathsf{key}},\mathcal{G}^{\mathsf{val}})$ is $L_{\mathsf{local}}$-local and relative (Assumption 3.2) with $L_{\mathsf{local}} = 2$. Moreover, coverage (Assumption 4.2) is satisfied with $\eta_\ell = \ell$.

However, note that the hypothesis $\hat{h} = h_{g_{00},g_1}$ achieves 0 expected loss on the distributions $\mathcal{P}_\ell$ for $\ell \leq L$, but $\Omega(1)$ expected loss on the distributions $\mathcal{P}_\ell$ for $\ell > L$.[12] Moreover, this example shows that we cannot even hope for an agnostic guarantee whereby $\hat{h}$ is close to the best-in-class hypothesis, since for any $\ell > L$, we in fact have[13]

$$E_{(\mathbf{X},\mathbf{Y}) \sim \mathcal{P}_\ell}[\mathcal{L}(\hat{h}(\mathbf{X}),\mathbf{Y})] \geq \mathbb{E}_{(\mathbf{X},\mathbf{Y}) \sim \mathcal{P}_\ell}[\mathcal{L}(h_{g_{01},g_1}(\mathbf{X}),\mathbf{Y})] + \Omega(1). \tag{15}$$

---

[12]Note that for $\mathbf{X} \sim \mathcal{P}_\ell$ for $\ell > L$, we will have that $g_{00}(S,\mathbf{X}_S) = -\infty$ for all $S$ with nonzero probability; here, recall from Definition 3.3 that $h_{g_{00},g_1}(\mathbf{X}) = \frac{1}{|\mathsf{Sets}_k([\ell])|}\sum_{S \in \mathsf{Sets}_k([\ell])} g_1(\mathbf{X}_S)$.

[13]Ensuring that (15) holds is the reason that we have $\mathcal{Q}_\ell^{\mathsf{pos}}$ put more mass on sets $S$ with $\max\{S\}-\min\{S\} = 2$ in (14).

**Removing coverage.** Finally, we show that Assumption 4.2 cannot be dropped. We consider the sparse attention class defined by the following $\mathcal{G}^{\text{key}}, \mathcal{G}^{\text{val}}$:

- $\mathcal{G}^{\text{key}} = \{g_{00}, g_{01}\}$, defined as follows:

$$g_{00}(S, \mathbf{x}) = \begin{cases} 0 & : \mathbf{x} = (2,2) \text{ and } \max\{S\} - \min\{S\} = 1 \\ -\infty & : \text{otherwise} \end{cases}$$

$$g_{01}(S, \mathbf{x}) = \begin{cases} 0 & : \mathbf{x} = (2,2) \text{ and } \max\{S\} - \min\{S\} \in \{1,2\} \\ -\infty & : \text{otherwise.} \end{cases}$$

- $\mathcal{G}^{\text{val}}$ consists of the single function $g_1 : \mathcal{V}^2 \to \mathcal{Y}$ defined as $g_1(2,2) = 1$ and $g_1(\mathbf{x}_1, \mathbf{x}_2) = 0$ for all $(\mathbf{x}_1, \mathbf{x}_2) \neq (2,2)$.

Fix an integer $L$. We define an ensemble $\mathcal{P}$ specified by the following $\mu, \mathcal{Q}_\ell^{\text{pos}}, \mathcal{Q}^{\text{voc}}, g^\star$:

- $\mu$ is the point mass on $\{1\}$.

- $\mathcal{Q}^{\text{voc}}$ is the point mass on $\{(2,2)\}$.

- $\mathcal{Q}_\ell^{\text{pos}}$ is uniform over the sets $\{1,2\}, \{2,3\}, ..., \{\ell-1, \ell\}$ if $\ell \leq L$, and uniform over the sets $\{1,3\}, \{2,4\}, ..., \{\ell-2, \ell\}$ if $\ell > L$.

- $g^\star = g_1 \in \mathcal{G}^{\text{val}}$.

It is immediate from the above definitions that $\mathcal{H}^{\text{attn}} = \mathcal{H}^{\text{attn}}(\mathcal{G}^{\text{key}}, \mathcal{G}^{\text{val}})$ is $L_{\text{local}}$-local and relative (Assumption 3.2) with $L_{\text{local}} = 2$. Moreover, realizability (Assumption 4.1) is satisfied by noting that $h_{g_{01}, g_1}$ has 0 loss.

However, note that the hypothesis $\hat{h} = h_{g_{00}, g_1}$ achieves 0 expected loss on the distributions $\mathcal{P}_\ell$ for $\ell \leq L$, but $\Omega(1)$ expected loss on the distribution $\mathcal{P}_\ell$ for $\ell > L$. $\qquad \square$

## D. Theoretical analysis for position coupling

In this section we formally state and prove Proposition 4.4. We begin with some notations. Given a set $\Omega \subset \mathbb{N}$ and $i \in \Omega$, we let $\text{rank}_\Omega(i)$ denote the rank of $i$ in $\Omega$, i.e., the position of $i$ in $\Omega$ when the elements of $\Omega$ are sorted. Next for $S \in \text{Sets}_k([\ell])$, $S = (i_1, ..., i_k)$, and $\psi_\ell : [\ell] \to [\ell]$, we let $\psi_\ell(S) \in \text{Sets}_k([\ell])$ to denote the set $\{\psi_\ell(i) : i \in S\}$.

First, we discuss some additional motivation for Proposition 4.4.

**Remark D.1** (Limitations of Assumption 3.2)**.** The combination of Assumptions 3.2 and 4.1 leads to the following restriction on the $k$-sparse distribution ensemble $\mathcal{P}$: in typical examples (modulo some degenerate ones where, e.g., all functions in $\mathcal{G}^{\text{val}}$ are constant), in order to satisfy realizability (Assumption 4.1), we will need the low-loss hypothesis $h^\star \in \mathcal{H}^{\text{attn}}(\mathcal{G}^{\text{key}}, \mathcal{G}^{\text{val}})$ to be of the form $h^\star = h_{g_0^\star, g_1^\star}$ for some $g_0^\star \in \mathcal{G}^{\text{key}}$ which "selects out" the planted set $S^\star$ and $g_1^\star \in \mathcal{G}^{\text{val}}$ which correctly evaluates the label $\mathbf{Y}$ given $\mathbf{X}_{S^\star}$; formally, for each $\ell \in \mathbb{N}$, with high probability under $(\mathbf{X}, S^\star, \mathbf{Y}) \sim \mathcal{P}_\ell$,

$$g_0^\star(S^\star, \mathbf{X}_{S^\star}) > -\infty, \quad g^\star(S, \mathbf{X}_S) = -\infty \;\; \forall S \neq S^\star,$$

and $g_1^\star(\mathbf{X}_{S^\star}) = \mathbf{Y}$. (We formally call this property *strong realizability* in Assumption D.1.) But by Item 1 of Assumption 3.2, this means that $\max\{S^\star\} - \min\{S^\star\} \leq L_{\text{local}}$ with high probability over the draw from $\mathcal{P}_\ell$. Ideally, we would like to establish results for planted $k$-sparse ensembles $\mathcal{P}$ for which the planted set $S^\star$ is *not* local in this sense; this is accomplished by Proposition D.2.

We now formally define the notion of a local positional coupling, which expands out the technical details omitted in the informal version in Definition 4.1.

**Definition D.2** (Local position coupling; Formal version of Definition 4.1)**.** Fix a distribution ensemble $\mathcal{P}$ with $k$-sparse planted correlations per Definition 3.2 (defined by $\mu, \mathcal{Q}^{\text{pos}}, \mathcal{Q}^{\text{voc}}$). A $L_{\text{local}}$-*local position coupling* of $\mathcal{P}$ is defined by, for each $\ell \in \mathbb{N}$, a joint distribution $\mathcal{Q}^{\text{pos}-\text{c}}$ over $S^\star \in \text{Sets}_k([\ell])$ and a mapping $\psi_\ell : [\ell] \to [\ell]$ so that the marginal of $S^\star$ under $\mathcal{Q}^{\text{pos}-\text{c}}$ is $\mathcal{Q}_\ell^{\text{pos}}$ and with probability 1 under the draw of $(S^\star, \psi_\ell) \sim \mathcal{Q}^{\text{pos}-\text{c}}$:

1. $\max\{\psi_\ell(S^\star)\} - \min\{\psi_\ell(S^\star)\} \leq L_{\text{local}}$.

2. For each $i \notin S^\star$, $|\psi_\ell^{-1}(\psi_\ell(i))| = 1$. (I.e., *Indices not in $S^\star$ are not coupled.*)

3. For some fixed $k'$ (independent of $\ell$), $|\psi_\ell(S^\star)| = k'$ and the distribution of the tuple $(\{\mathrm{rank}_{\psi_\ell^{-1}(S^\star)}(j) : j \in \psi_\ell^{-1}(i)\})_{i \in \psi_\ell(S^\star)}$ does not depend on $\ell$.[14]

**Position coupling operation on functional attention classes.** We say that a mapping $g_0 : (\mathbb{N} \times \mathcal{V})^k \to \mathbb{R} \cup \{-\infty\}$ (e.g., an element of $\mathcal{G}^{\mathsf{key}}$ is *position-independent* if $g_0(S, \mathbf{X})$ depends only on $\mathbf{X}$ (and not $S$). For such $g_0$, we will write $g_0 : \mathcal{V}^k \to \mathbb{R} \cup \{-\infty\}$. We say that $\mathcal{G}^{\mathsf{key}}$ is *position-independent* if all $g_0 \in \mathcal{G}^{\mathsf{key}}$ are position-independent.

We define a new vocabulary $\bar{\mathcal{V}}_k := (\mathcal{V} \sqcup [k])^{\leq k}$, and "position-coupled" versions, $\mathsf{PC}[g_0] : (\mathbb{N} \times \bar{\mathcal{V}}_k)^k \to \mathbb{R} \cup \{-\infty\}$ and $\mathsf{PC}[g_1] : \bar{\mathcal{V}}_k^k \to \mathcal{Y}$ of mappings $g_0 : \mathcal{V}^k \to \mathbb{R} \cup \{-\infty\}, g_1 : \mathcal{V}^k \to \mathcal{Y}$, as follows. First, given $\mathbf{X} \in \bar{\mathcal{V}}_k^k$, let

$$\mathrm{expand}(\mathbf{X}) \in \mathcal{V}^\star$$

be defined as follows: for each $i \in [k]$, we can write $\mathbf{X}_i = ((j_{i1}, \mathbf{Z}_{i1}), \dots, (j_{in}, \mathbf{Z}_{in}))$ for some $n \geq 0$, $j_{i1}, \dots, j_{in} \in [\ell]$ and $\mathbf{Z}_{i1} \dots, \mathbf{Z}_{in} \in \mathcal{V}$. Then we let $\mathrm{expand}(S, \mathbf{X})$ be the tuple consisting of all pairs $(j_{i1}, \mathbf{Z}_{i1}), \dots, (j_{in}, \mathbf{Z}_{in})$ across all $i \in [k]$, where the indices $j_{i1}, \dots, j_{in}$ are sorted in increasing order across all $i$ and all duplicates in the indices are removed. We next define

$$\mathsf{PC}[g_0](S, \mathbf{X}) := \begin{cases} -\infty & : \max\{S\} - \min\{S\} > L_{\mathsf{local}} \\ -\infty & : \mathrm{expand}(\mathbf{X}) \notin \mathcal{V}^k \\ g_0(\mathrm{expand}(\mathbf{X})) & : \text{otherwise,} \end{cases} \tag{16}$$

so that $\mathsf{PC}[g_0]$ maps $(\mathbb{N} \times \bar{\mathcal{V}}_k)^k \to \mathbb{R} \cup \{-\infty\}$, and

$$\mathsf{PC}[g_1](\mathbf{X}) := \begin{cases} g_1(\mathrm{expand}(\mathbf{X})) & : \mathrm{expand}(S) \in \mathcal{V}^k \\ y_0 & : \text{otherwise,} \end{cases} \tag{17}$$

where $y_0 \in \mathcal{Y}$ is an arbitrary element; thus $\mathsf{PC}[g_1]$ maps $\bar{\mathcal{V}}_k^k \to \mathcal{Y}$. For a sparse functional attention class $\mathcal{H}^{\mathsf{attn}} = \mathcal{H}^{\mathsf{attn}}(\mathcal{G}^{\mathsf{key}}, \mathcal{G}^{\mathsf{val}})$, we define

$$\mathsf{PC}[\mathcal{H}^{\mathsf{attn}}] := \{h_{\mathsf{PC}[g_0], \mathsf{PC}[g_1]} : g_0 \in \mathcal{G}^{\mathsf{key}}, g_1 \in \mathcal{G}^{\mathsf{val}}\}.$$

Note that hypotheses in $\mathsf{PC}[\mathcal{H}^{\mathsf{attn}}]$ map $\bar{\mathcal{V}}_k^\ell \to \mathcal{Y}$.

**Position coupling operation on distribution ensembles.** Next, we define a "position coupling" operation which modifies a distribution ensemble equipped with a local position coupling (per Definition D.2) to produce sequences in $\bar{\mathcal{V}}_k$ with tokens coupled between positions.

Fix a distribution ensemble $(\mathcal{P}_\ell)_{\ell \in \mathbb{N}}$ with $k$-sparse planted correlations, which is specified by the tuple $(\mu, \mathcal{Q}_\ell^{\mathsf{pos}}, \mathcal{Q}^{\mathsf{voc}})$. Given a sequence $\mathbf{X} \in \mathcal{V}^\ell$, a set $S \in \mathrm{Sets}_k([\ell])$, and a function $\psi_\ell : [\ell] \to [\ell]$, we let $\mathsf{PC}[\mathbf{X}, S]$ be the distribution over sequences $\bar{\mathbf{X}} \in \bar{\mathcal{V}}_k^\ell$ defined as follows: for $i \in [\ell]$, the distribution of $\bar{\mathbf{X}}_i$ is given by

$$\bar{\mathbf{X}}_i \begin{cases} = ((\mathrm{rank}_{\psi_\ell^{-1}(S)}(j), \mathbf{X}_j))_{j \in \psi_\ell^{-1}(i)} & : i \in \psi_\ell(S) \\ = (I, \mathbf{X}_{\min(\psi_\ell^{-1}(i))}), I \sim \mathrm{Unif}([\ell]) & : i \notin \psi_\ell(S), \psi_\ell^{-1}(i) \neq \emptyset \\ \sim \mathrm{Unif}([\ell]) \times \mu & : \text{otherwise,} \end{cases} \tag{18}$$

In words, we are "rearranging" the tokens of $\mathbf{X}$ by grouping together (i.e., "coupling") tokens indexed by $S$ which are mapped by $\psi_\ell$ to the same index. Moreover, for positions $i$ not in $\psi_\ell(S)$, we take the token at the smallest position of $\psi_\ell^{-1}(i)$, and for positions $i$ not in the image of $\psi_\ell$, we sample $\bar{\mathbf{X}}_i$ from $\mathrm{Unif}([\ell]) \times \mu$. We let $\mathsf{PC}[\mathcal{P}_\ell]$ denote the distribution of $\mathsf{PC}[\mathbf{X}, S^\star]$ where $(\mathbf{X}, S^\star) \sim \mathcal{P}_\ell$.

**Strong realizability.** Finally, we introduce the following slight strengthening of realizability:

---

[14]For a set $\Omega \subset \mathbb{N}$ and $i \in \Omega$, $\mathrm{rank}_\Omega(i)$ is the position of $i$ in $\Omega$ when the elements of $\Omega$ are sorted in increasing order; see Appendix D. Item 3 is a technical condition which is satisfied by most practical position coupling schemes which have been proposed.

**Assumption D.1** (Strong realizability)**.** Write $\mathcal{H}^{\mathsf{attn}} = \mathcal{H}^{\mathsf{attn}}(\mathcal{G}^{\mathsf{key}}, \mathcal{G}^{\mathsf{val}})$. We say that an ensemble $\mathcal{P}$ is $\delta$-*strongly approximately* $\mathcal{H}^{\mathsf{attn}}$-*realizable* if there are $g_0^\star \in \mathcal{G}^{\mathsf{key}}, g_1^\star \in \mathcal{G}^{\mathsf{val}}$, for each $\ell \in \mathbb{N}$, with probability $1 - \delta$ under $(\mathbf{X}, S^\star, \mathbf{Y}) \sim \mathcal{P}_\ell$,

$$g_0^\star(S^\star, \mathbf{X}_{S^\star}) > -\infty, \quad g_0^\star(S, \mathbf{X}_S) = -\infty \ \ \forall S \neq S^\star,$$

and $g_1^\star(\mathbf{X}_{S^\star}) = \mathbf{Y}$.

Using the fact that the diameter of $\mathbf{Y}$ is at most 1, it is straightforward that $\delta$-strong approximate realizability implies $\delta$-approximate realizability.

**Position coupling removes the need for locality.**    The below proposition confirms that applying the position-coupling operation to a distribution ensemble $\mathcal{P}_\ell$ with $k$-sparse planted correlations (Definition 3.2), to yield $\mathsf{PC}[\mathcal{P}_\ell]$, and applying the position coupling operation PC to a sparse functional attention class $\mathcal{H}^{\mathsf{attn}}$, to yield $\mathsf{PC}[\mathcal{H}^{\mathsf{attn}}]$, can effectively remove the requirement for locality to hold (per Assumption 3.2) in order to obtain provable length generalization:

**Proposition D.2** (Length generalization without locality from position coupling)**.** *Suppose that $\mathcal{P} = (\mathcal{P}_\ell)_{\ell \in \mathbb{N}}$ is an ensemble with $k$-sparse planted correlations (Definition 3.2) defined by distributions $(\mu, \mathcal{Q}_\ell^{\mathsf{pos}}, \mathcal{Q}^{\mathsf{voc}})$, and for some $L_{\mathsf{local}} \in \mathbb{N}$ is amenable to $L_{\mathsf{local}}$-position coupling (Definition D.2). Moreover suppose that the distribution of $\psi_\ell(S^\star)$ for $(S^\star, \psi_\ell) \sim \mathcal{Q}_\ell^{\mathsf{pos-c}}$ satisfies the coverage assumption in Assumption 4.2 for some values of $\eta_\ell > 0$, $\ell \in \mathbb{N}$.*

*Then for any sparse functional attention class $\mathcal{H}^{\mathsf{attn}} = \mathcal{H}^{\mathsf{attn}}(\mathcal{G}^{\mathsf{key}}, \mathcal{G}^{\mathsf{val}})$ for which $\mathcal{G}^{\mathsf{key}}$ is position-independent and $\mathcal{P}$ is $\delta$-approximately strongly $\mathcal{H}^{\mathsf{attn}}$-realizable, and any integers $L, \bar{L} \in [L_{\max}]$ for which $L_{\mathsf{local}} \mid \bar{L} - L$ and $L \geq 4L_{\mathsf{local}}$, the class $\mathsf{PC}[\mathcal{H}^{\mathsf{attn}}]$ achieves $(L, \bar{L}, \eta_L \eta_{\bar{L}} \cdot L\bar{L}^2 \cdot \delta)$-length generalization with respect to the ensemble $\mathsf{PC}[\mathcal{P}]$.*

*Proof of Proposition D.2.* We will verify that the sparse functional attention class $\mathsf{PC}[\mathcal{H}^{\mathsf{attn}}]$ together with the ensemble $\mathsf{PC}_{\mu, \psi}[\mathcal{P}]$ satisfy the requirements of Theorem 4.3 with respect to the vocabulary $\bar{\mathcal{V}}$ defined above.

First, it is straightforward to see that $\mathsf{PC}_{\mu, \psi}[\mathcal{P}]$ is itself a $k$-sparse planted correlations distribution ensemble: for length $\ell$, the planted set is given by $\psi_\ell(S^\star)$ for $S^\star \sim \mathcal{P}_\ell$, and for all $i \notin S^\star$, the distribution of $\bar{\mathbf{X}}_i$ (for $\bar{\mathbf{X}} \sim \mathsf{PC}[\mathcal{P}_\ell]$) is an independent sample from $\mathsf{Unif}([\ell]) \times \mu$. (Here we have used the second and third cases in (18) together with Item 2 of Definition D.2.) We have additionally used Item 3 of Definition D.2, which ensures that the distribution of $\bar{\mathbf{X}}_{\psi_\ell(S^\star)}$ does not depend on $\ell$ (recall from Definition 3.2 that $\mathcal{Q}^{\mathsf{voc}}$ cannot depend on $\ell$).

Next we verify each of the three assumptions made in Theorem 4.3:

- Assumption 3.2 is satisfied by the first line of (16) (which ensures $L_{\mathsf{local}}$-locality) and since $\mathsf{PC}[g_0](S, \mathbf{X})$ does not depend on $S$ for sets $S$ satisfying $\max\{S\} - \min\{S\} \leq L_{\mathsf{local}}$ (again by (16)).

- To show that Assumption 4.1 is satisfied in the sense that $\mathsf{PC}[\mathcal{P}]$ is $\delta$-approximately $\mathsf{PC}[\mathcal{H}^{\mathsf{attn}}]$-realizable, we use that $\mathcal{P}$ is $\delta$-strongly approximately $\mathcal{H}^{\mathsf{attn}}$-realizable. In particular, choose $g_0^\star \in \mathcal{G}^{\mathsf{key}}, g_1^\star \in \mathcal{G}^{\mathsf{val}}$ per Assumption D.1. Consider a sample $(\bar{\mathbf{X}}^\star, \bar{S}^\star) \sim \mathsf{PC}[\mathcal{P}_\ell]$. We can write $\bar{S}^\star = \psi_\ell(S^\star)$ and $\bar{\mathbf{X}}$ as in (18) for a sample $(\mathbf{X}, S^\star, \psi_\ell) \sim \mathcal{P}_\ell$. Thus, with probability $1 - \delta$ under $(\bar{\mathbf{X}}, \bar{S}^\star) \sim \mathsf{PC}[\mathcal{P}_\ell]$, we have from Eqs. (16) and (18) that

$$\mathsf{PC}[g_0^\star](\bar{S}^\star, \bar{\mathbf{X}}_{\bar{S}^\star}) = g_0^\star(S^\star, \mathbf{X}_{S^\star}) > -\infty.$$

  Moreover, in order to have $\mathsf{PC}[g_0^\star](S, \bar{\mathbf{X}}_S) > -\infty$ for some $S \in \mathsf{Sets}_k([\ell])$, we need (under the $1 - \delta$ probability event above) that $\mathsf{expand}(\bar{\mathbf{X}}_S) = \mathbf{X}_{S^\star}$; here we have used that $g_0^\star \in \mathcal{G}^{\mathsf{key}}$ is position-independent. For any such set $S$, it holds also that $\mathsf{PC}[g_1^\star](\bar{\mathbf{X}}_S) = g_1^\star(\mathsf{expand}(\bar{\mathbf{X}}_S)) = g_1^\star(\mathbf{X}_{S^\star})$, which is equal to the label $\bar{\mathbf{Y}}$ on this event.

- Assumption 4.2 is satisfied on account of the assumption from the proposition statement on the distribution of $\psi_\ell(S^\star)$ for $(S^\star, \psi_\ell) \sim \mathcal{Q}_\ell^{\mathsf{pos-c}}$.

$\square$

**Remark D.3** (Theoretical justification for PoSE)**.** It is natural to wonder if our theoretical framework allows us to justify other techniques used to induce length generalization, such as PoSE (Appendix B.3), in a sense akin to Proposition 4.4. At a high level, PoSE is adjusting the distribution of position IDs during training, so that IDs always seen farther apart than the training context window at test time may nevertheless be observed in the same training instance. In other words, a model trained as such should "interpret" a greater range of sets $S$ as satisfying the locality requirement of Assumption 3.2. Thus, we conjecture that this adjustment allows us to remove the locality requirement in Assumption 3.2 as well; we leave a formal proof of this fact as an intriguing direction for future work.

# E. Additional experimental details: synthetic data experiments

For our experiments with synthetic data (Sections 5.1 and 5.2), we used the GPT-NeoX decoder-only transformer model with a causal attention mask. The transformer had 12 layers, 16 heads, and an embedding dimension of 1024. We used the AdamW optimizer with learning rate $5 \cdot 10^{-5}$ and weight decay parameter equal to 0.1; moreover, the experiments use a linear learning rate scheduler with 300 warmup steps. All shaded areas in figures represent 95% confidence intervals, computed with respect to multiple training runs (the precise number of training runs for each experiment is noted in the following subsections).

**Remark E.1.** One might wonder why we emphasize Item 1 but not Item 2 of Assumption 3.2 in takeaway (T2). In fact, the conceptual message of Item 2, namely that position IDs only influence attention scores by their *relative* information (i.e., the difference between different positions) is already captured by the fact that it is common to use *relative positional embeddings* and variants (e.g., RoPE (Su et al., 2021), FIRE (Li et al., 2024)) in many open-source transformer architectures. Due to the success of such embeddings, in this sense the constraint imposed by Item 2 can "come for free".

## E.1. Sparse parity (Section 5.1)

E.1.1. HYPERPARAMETERS AND DATA FORMAT.

**Hyperparameters.** The hyperparameters specific to our sparse parity experiments are shown in Table 1.

| Parameter | Value |
|---|---|
| Minimum training length | 20 |
| Maximum training length | 50 |
| Minimum testing length | 20 |
| Maximum testing length | 500 |
| Training batch size | 64 |
| Number of training steps | 50,000 |
| Number of testing points | 512 |
| Vocabulary size $N$ | 20 |
| Maximum position ID ($L_{\mathsf{max}}$) | 510 |
| Positional embedding | RoPE |
| Number of training runs | 1 |

Table 1: Hyperparameters for sparse parity experiments (Section 5.1).

**Data format.** For some positive integer $N$, we consider an alphabet of $\mathcal{V} = [N] \cup \{\mathsf{t}_0, \mathsf{t}_1, \texttt{<BOS>}, \texttt{<SEP>}\}$. Given integers $k, \ell$ satisfying $k < \ell$, we consider the task of evaluating the parity of a particular set of $k$ out of $\ell$ tokens at even-numbered positions. The particular set of $k$ tokens is defined as those tokens for which the directly preceding token (which must belong to $[N]$) is $\leq N/2$. We remark that this task makes sense even for $N = 2$.

Formally, the task is defined as follows. For each $\ell, k \in \mathbb{N}$, we let $\mathcal{D}_{\ell,k}^{\mathsf{sp}}$ denote the distribution over sequences of the form

$$\texttt{<BOS>}, J_1, \mathsf{t}_{b_1}, ..., J_\ell, \mathsf{t}_{b_\ell}, \texttt{<SEP>}, \mathsf{t}_{b^\star} \tag{19}$$

where $(J_1, ..., J_\ell) \in [N]^\ell$ is chosen uniformly subject to the constraint that exactly $k$ of the values $J_1, ..., J_\ell$ are $\leq N/2$ and the remaining $\ell - k$ values are $> N/2$. Moreover, $b_1, ..., b_\ell \sim \mathrm{Unif}\{0,1\}$, and $b^\star = \bigoplus_{i: J_i \leq N/2} b_i$. We consider the task of predicting the final token $\mathsf{t}_{b^\star}$.

This task fits into the framework discussed in Section 3 as follows: $\mathcal{P}_\ell$ is the distribution over tuples $(\mathbf{X}, \mathbf{Y})$ distributed as in (19) where $\mathbf{X}$ consists of all tokens except the final one $\mathsf{t}_{b^\star}$, and $\mathbf{Y} = \mathsf{t}_{b^\star}$.[15] It is straightforward to verify that, under the additional assumption that the $k$ values of $i$ for which $J_i \leq N/2$ are all within $L_{\mathsf{local}}$ of each other (for some choice of $L_{\mathsf{local}}$), then Assumption 4.2 is satisfied for $\eta_\ell = O(\ell)$ as long as $\ell \geq \Omega(L_{\mathsf{local}})$, and Assumptions 3.2 and 4.1 are both satisfied for natural choices of the sparse functional attention class $\mathcal{H}^{\mathsf{attn}}$.

---

[15]Technically, the length of $\mathbf{X}$ is $2\ell$ instead of $\ell$, but this discrepancy is immaterial.

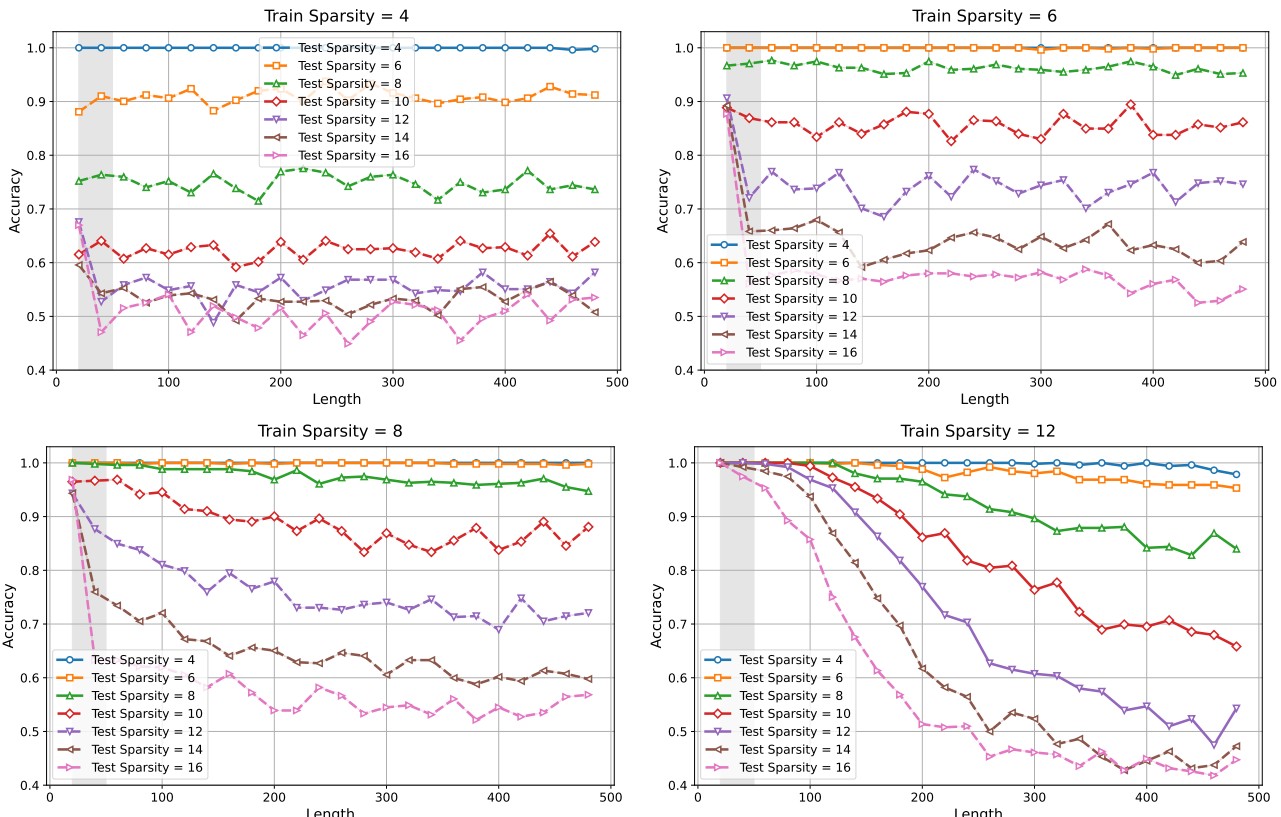

Figure 3: Length generalization for the sparse parity task for values $K_{\text{train}} \in \{4,6,8,12\}$ (see Figure 1a for $K_{\text{train}} = 10$).

### E.1.2. ADDITIONAL RESULTS

The analogous plots to Figure 1a for $K_{\text{train}} \in \{4,6,8,12\}$ are shown in Figure 3. The same patterns for $K_{\text{train}} = 10$ are evident for each of these other values of $K_{\text{train}}$ (except that the values of $K_{\text{train}} \in \{4,6\}$ are sufficiently small so that there is essentially perfect length generalization for each value of $k \leq K_{\text{train}}$).

### E.2. Parity with scratchpad (Section 5.2.1)

#### E.2.1. HYPERARAMETERS AND DATA FORMAT

**Hyperarameters.** The hyperparameters specific to our parity-with-scratchpad experiments are shown in Table 2.

**Data format.** For each length $\ell \in \mathbb{N}$, we train using data of the following format, which computes the parity of $\ell$ bits by successively computing the parity of the first $i$ bits for each $i = 1,2,...,\ell$:

$$\texttt{<BOS>},b_1,...,b_\ell,\texttt{<SEP>},b'_1,...,b'_\ell \tag{20}$$

where $b_1,...,b_\ell \sim \text{Unif}\{0,1\}$ are independent and $b'_i = \bigoplus_{j=1}^{i} b_j$ for $i \in [\ell]$. The model is trained to predict all tokens after the $\texttt{<SEP>}$ (namely, the scratchpad). For position coupling, we use position IDs of $0,1,...,\ell,0,1,...,\ell$ (which are shifted by a random offset during training as discussed above). Note that in this task, the position IDs used in position coupling in fact *are* determined solely by the length of the sequence.

We train on $\ell$ for which the combined length of input and scratchpad satisfies $2\ell \in [10,40]$ and test for $\ell$ satisfying $2\ell \in [10,200]$.

#### E.2.2. ADDITIONAL RESULTS

In Figure 4, we show the length generalization behavior for our experimental setup with modifications that (a) remove Predictive Position Coupling and simply use absolute position embeddings with a random shift during training time, and (b) use RoPE with

PoSE. Both of these modifications significantly harm length generalization behavior.

| Parameter | Value |
|---|---|
| Minimum training length | 10 |
| Maximum training length | 40 |
| Minimum testing length | 10 |
| Maximum testing length | 200 |
| Training batch size | 64 |
| Number of training steps | 50,000 |
| Number of testing points | 192 |
| Maximum position ID ($L_{\max}$) | 210 |
| Positional embedding | Absolute Position Embeddings (learned) |
| Number of training runs | 10[16] |

Table 2: Hyperparameters for parity-with-scratchpad experiments (Section 5.2.1).

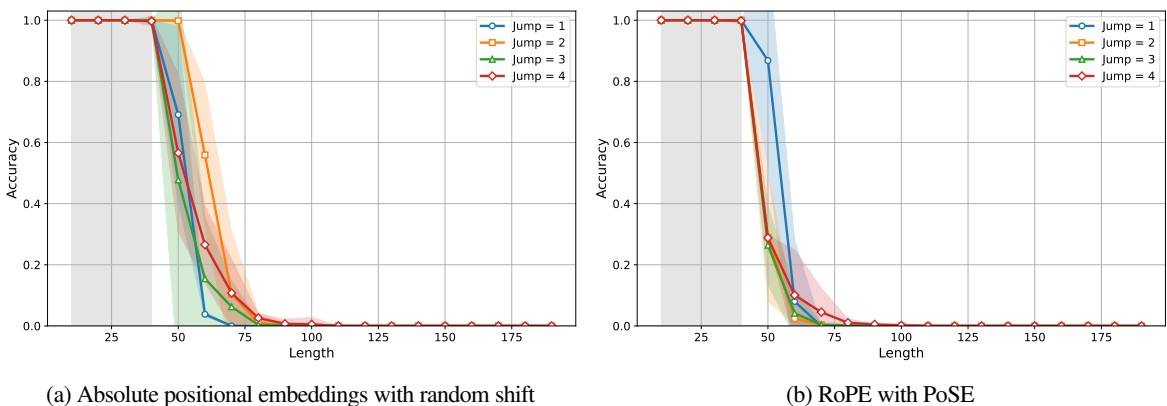

(a) Absolute positional embeddings with random shift
(b) RoPE with PoSE

Figure 4: Length generalization figure for parity-with-scratchpad experiments (Section 5.2.1), with modifications that replace Predictive Position Coupling. Length generalization behavior is significantly worse than that in Figure 1b.

### E.3. Variable assignment with scratchpad (Section 5.2.2)

| Parameter | Value |
|---|---|
| Minimum training length | 20 |
| Maximum training length | 60 |
| Minimum testing length | 20 |
| Maximum testing length | 200 |
| Training batch size | 32 |
| Number of training steps | 50,000 |
| Number of testing points | 192 |
| Vocabulary size (number of variables) | 200 |
| Maximum position ID ($L_{\max}$) | 230 |
| Positional embedding | Absolute positional embedding (learned) |
| Number of training runs | 3[17] |

Table 3: Hyperparameters for variable assignment-with-scratchpad experiments (Section 5.2.2).

---

[16]For the ablation studies in Figure 4, we only used 3 training runs.
[17]The ablation study in Figure 5b uses 10 training runs.

E.3.1. HYPERPARAMETERS AND DATA FORMAT

**Hyperparameters.** The hyperparameters specific to our variable assignment-with-scratchpad experiments are shown in Table 3.

**Data format.** We now describe in detail the distribution of data for the "variable assignment with scratchpad" problem discussed in Section 5.2.2. To sample a problem instance, we first draw some number $w$ of *chains*, where a *chain* is a sequence of $d$ variable assignments of the form: $v_1 \leftarrow v_2; v_2 \leftarrow v_3; \cdots; v_d \leftarrow v_{d+1}$. We sample uniformly at random a sequence of $w$ such chains with distinct variable IDs, where chain $j$ is denoted $v_{j,1} \leftarrow v_{j,2}; v_{j,2} \leftarrow v_{j,3}; \cdots; v_{j,d} \leftarrow v_{j,d+1}$. We interleave these chains randomly (but keep the order within each chain preserved). We let $v_{\mathsf{init}} = v_{1,1}$, and begin the input sequence with $v_{1,1}$ (so that it is possible for the model to determine which chain to follow). Finally, the sequence concludes with a scratchpad of the form $v_{1,1} \leftarrow v_{1,2}; v_{1,2} \leftarrow v_{1,3}; \cdots; v_{1,d} \leftarrow v_{1,d+1}$, evaluating the variable assignments along the correct chain. Overall, an example sequence consists of the following token IDs (here, $\leftarrow$, <SEP$_1$>, <SEP$_2$> are separator tokens):

$$
\begin{aligned}
&\texttt{<BOS>}, v_{1,1}, \texttt{<SEP}_1\texttt{>}, v_{\pi(1),\sigma(1)}, \leftarrow, v_{\pi(1),\sigma(1)+1}, v_{\pi(2),\sigma(2)}, \leftarrow, v_{\pi(2),\sigma(2)+1}, ..., \texttt{<SEP}_2\texttt{>},\\
&v_{1,1}, \leftarrow, v_{1,2}, ..., v_{1,d}, \leftarrow, v_{1,d+1}.
\end{aligned}
\tag{21}
$$

Here we have used $\pi(1), \pi(2), ... \in [w]$ to denote the chain index corresponding to each step in the input sequence, and $\sigma(1), \sigma(2), ... \in [d]$ to denote the position within its chain corresponding to each step in the input sequence. As an example with $v = 2$ and $w = 2$, we might have the sequence:

$$
\texttt{<BOS>}, v_{1,1}, \texttt{<SEP}_1\texttt{>}, v_{2,1}, \leftarrow, v_{2,2}, v_{1,1} \leftarrow v_{1,2}, v_{1,2}, \leftarrow v_{1,3}, v_{2,2} \leftarrow v_{2,3}, \texttt{<SEP}_2\texttt{>}, v_{1,1}, \leftarrow, v_{1,2}, v_{1,2}, \leftarrow, v_{1,3}.
$$

The model is trained to predict all tokens after the <SEP$_2$>. For a fixed depth $d$ and desired length $\ell$ of the input, we choose $w$ as large as possible so that the length of the sequence between <SEP$_1$> and <SEP$_2$> is at most $\ell$.

For position coupling, the first part of the example (before <SEP$_2$>) uses the true position IDs $1, 2, 3, ..., \ell$, where $\ell$ denotes the number of tokens before <SEP$_2$>. The final part of the example (after <SEP$_2$>) uses position IDs $I_1, I_1+1, I_1+2, I_2, I_2+1, I_2+2, ...,$ where for $j \in [d]$, $I_j$ denotes the position ID assigned to $v_{1,j}$ in the first part of the example. Summarizing, the position IDs corresponding to the example (21) are:

$$
0, 1, 2, ..., \ell, 0, I_1, I_1+1, I_1+2, ..., I_d, I_d+1, I_d+2.
$$

Note that, because $I_1, I_2, ..., I_d$ depend on the (random) functions $\pi, \sigma$, it is crucial that the model learns to predict these coupled position IDs, so that they can be passed as the next position ID.

E.3.2. ADDITIONAL RESULTS

Figure 5 shows the length generalization behavior for our experimental setup with modifications that (a) remove Predictive Position Coupling and use absolute positional embeddings with a random shift during training time, (b) use RoPE with PoSE (keeping the scratchpad), and (c) use RoPE with PoSE and remove the scratchpad. Notice that Predictive Position Coupling (Figure 1c) greatly outperforms all 3 of these modifications.

# F. Additional experimental details: Natural Language Experiments

**Additional experimental details.** Hyperparameters used for our training procedure for the model $\hat{h}_{\mathsf{short}}$, discussed in Section 5.3. The model $\hat{h}_{\mathsf{short}}$ with context length $L = 64$ discussed in Section 5.3 was trained for 34000 steps without PoSE, and 15000 steps with PoSE. We additionally trained a model $\hat{h}_{\mathsf{long}}$ with context length $\bar{L} = 128$ for 48000 steps, whose hyperparameters are identical to those of $\hat{h}_{\mathsf{short}}$ except the batch and microbatch sizes are halved. Altogether, both models $\hat{h}_{\mathsf{short}}, \hat{h}_{\mathsf{long}}$ were trained on roughly 25B tokens, and each has 1.3B parameters.

**Formal setup for results.** Let $\hat{h}_{\mathsf{short}}(\mathbf{X}_i \mid \mathbf{X}_{1:i-1})$ denote the probability $\hat{h}_{\mathsf{short}}$ assigns to token $\mathbf{X}_i$ given the context $\mathbf{X}_{1:i-1}$. For each test example $\mathbf{X} = (\mathbf{X}_1, ..., \mathbf{X}_{\bar{L}})$, we first computed a set of $k = 5$ "influential" tokens for predicting $\mathbf{X}_{\bar{L}}$, chosen amongst the first $\bar{L} - L$ tokens, by finding the $k$ values of $j \in [\bar{L} - L - 1]$ which minimize

$$
\hat{h}_{\mathsf{short}}(\mathbf{X}_{\bar{L}} \mid \mathbf{X}_{1:j-1}, \mathbf{X}_{j+1:\bar{L}-1}),
\tag{22}
$$

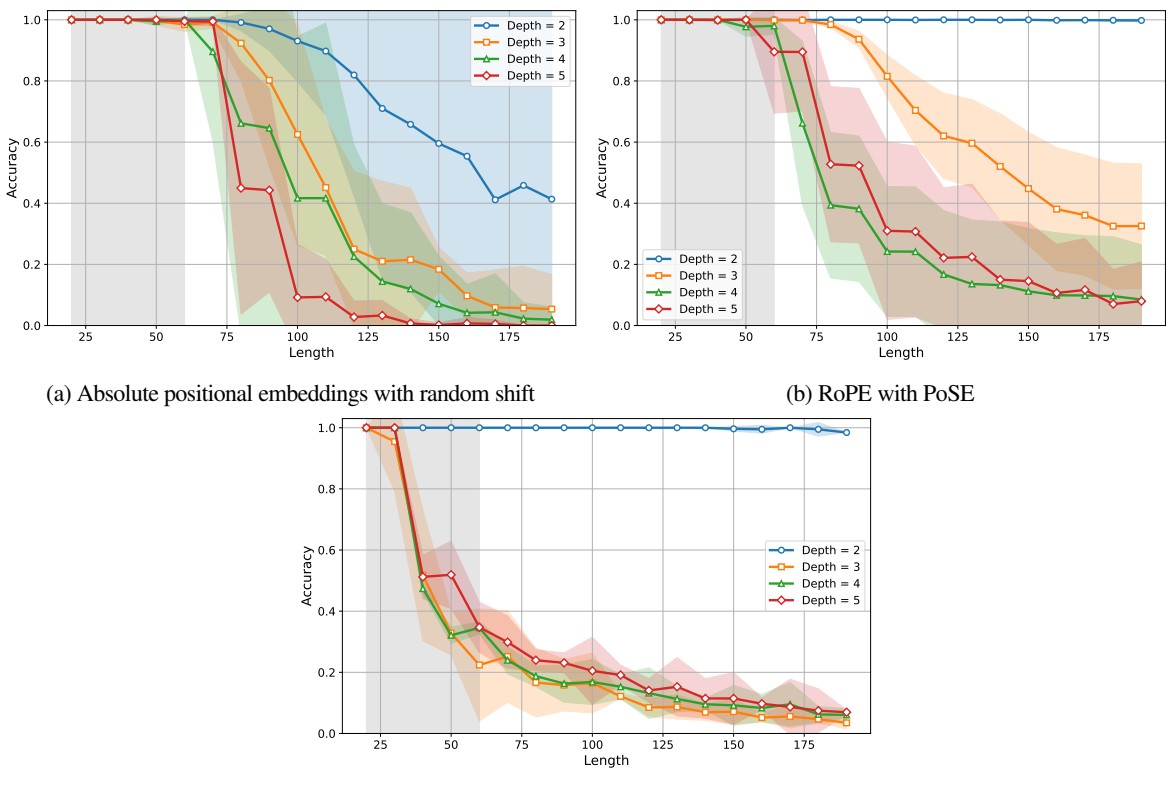

(a) Absolute positional embeddings with random shift  (b) RoPE with PoSE

(c) RoPE with PoSE & No Scratchpad

Figure 5: Length generalization figure for variable assignment experiments (Section 5.2.1). Figs. 5a to 5c show length generalization behavior with modifications that replace Predictive Position Coupling; length generalization is significantly worse than that in Figure 1c. Figs. 5a and 5c report the full-string accuracy on the scratchpad; Figure 5b (which does not use a scratchpad) reports the accuracy of the model at predicting the answer token. Moreover, consistent with takeaway (T1) it appears that, even when restricting to RoPE with PoSE, having a scratchpad is superior to not having one (even when accuracy in the scratchpad case is measured by full-string correctness, as in Figure 5c), though there is significant variance between different training runs.

i.e., where token $\mathbf{X}_j$ is masked out in all attention computations. Let $J_1,...,J_k$ denote these $k$ tokens.[18] We then compute the following three negative log-likelihoods:

$$\mathcal{L}_{\mathsf{short}} := -\log \hat{h}_{\mathsf{short}}(\mathbf{X}_{\bar{L}} \,|\, \mathbf{X}_{\bar{L}-L:\bar{L}-1})$$
$$\mathcal{L}_{\mathsf{long}} := -\log \hat{h}_{\mathsf{short}}(\mathbf{X}_{\bar{L}} \,|\, \mathbf{X}_{1:\bar{L}-1})$$
$$\mathcal{L}_{\mathsf{short,sparse}} := -\log \hat{h}_{\mathsf{short}}(\mathbf{X}_{\bar{L}} \,|\, \mathbf{X}_{J_{1:k}}, \mathbf{X}_{\bar{L}-L:\bar{L}-1}),$$

where to compute $\mathcal{L}_{\mathsf{short,sparse}}$ we mask out all tokens in the first $\bar{L} - L$ tokens except those at positions $J_1,...,J_k$. In words, $\mathcal{L}_{\mathsf{long}}, \mathcal{L}_{\mathsf{short}}, \mathcal{L}_{\mathsf{short,sparse}}$ denote the cross-entropy losses for predicting $\mathbf{X}_{\bar{L}}$ when the (a) full context $\mathbf{X}_{1:\bar{L}-1}$ is used, (b) the $L$ most recent tokens $\mathbf{X}_{\bar{L}-L:\bar{L}-1}$ are used, and (c) the $L$ most recent tokens as well as $\mathbf{X}_{J_{1:k}}$ are used, respectively. We illustrate the quantities $\mathcal{L}_{\mathsf{short}}, \mathcal{L}_{\mathsf{long}}, \mathcal{L}_{\mathsf{short,sparse}}$ for a particular choice of $L, \bar{L}, J_{1:k}$ in Figure 6.

**Results & discussion.** In Figure 2, we plotted the pairs $(\mathcal{L}_{\mathsf{short}} - \mathcal{L}_{\mathsf{long}}, \mathcal{L}_{\mathsf{short}} - \mathcal{L}_{\mathsf{short,sparse}})$ for each example $\mathbf{X}$. As can be seen, the quantity $\mathcal{L}_{\mathsf{short}} - \mathcal{L}_{\mathsf{long}}$, which may be interpreted as the *amount the model $\hat{h}_{\mathsf{short}}$ uses tokens $\mathbf{X}_{1:\bar{L}-L-1}$ to improve the prediction of $\mathbf{X}_{\bar{L}}$*, is roughly equal (up to noise) to $\mathcal{L}_{\mathsf{short}} - \mathcal{L}_{\mathsf{short,sparse}}$, which may be interpreted as the *amount the model uses tokens $\mathbf{X}_{J_1},...,\mathbf{X}_{J_k}$ to improve the prediction of $\mathbf{X}_{\bar{L}}$*. This provides evidence for the hypothesis that the model's prediction of the $\bar{L}$th token is sparse in its dependence on tokens "far in the past", and is thus consistent with (T1) which predicts that such sparsity allows the model $\hat{h}_{\mathsf{short}}$ to length-generalize.

It is natural to wonder whether the behavior seen in Figure 2 occurs even absent the use of methods such as PoSE to extend the context

---

[18]This is essentially equivalent to the "leave-one-out" baseline in Cohen-Wang et al. (2024).

| Hyperparameter | $\hat{h}_{\text{short}}$ | $\hat{h}_{\text{long}}$ |
|---|---|---|
| Model Dimension | 2048 | 2048 |
| MLP Hidden Size | 8192 | 8192 |
| Number of Heads | 32 | 32 |
| Number of Layers | 24 | 24 |
| Position Embedding | RoPE (w/ PoSE) | RoPE |
| Activation Type | GeLU | GeLU |
| Vocabulary Size | 32000 | 32000 |
| Tokenizer | Llama-2-7b | Llama-2-7b |
| Batch Size | 8192 | 4096 |
| Microbatch (per-device) size | 256 | 128 |
| Training Context Length | 64 | 128 |
| Number of training steps | 49000 | 48000 |
| Optimizer | AdamW (w/o weight decay) | AdamW (w/o weight decay) |
| Learning Rate | $10^{-3}$ | $10^{-3}$ |

Table 4: Hyperparameters for our natural language modeling experiments (based off of OLMo codebase with C4 dataset).

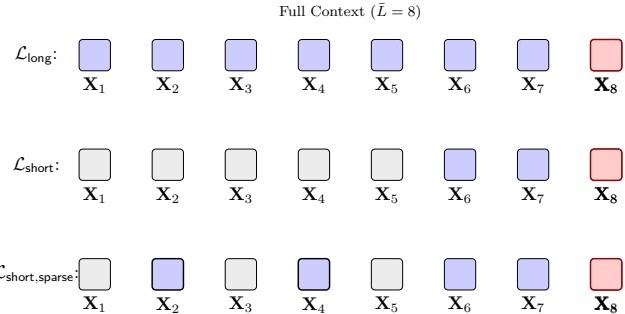

Figure 6: Illustration of the computation of $\mathcal{L}_{\text{short}}, \mathcal{L}_{\text{long}}, \mathcal{L}_{\text{short,sparse}}$ with $\bar{L}=8, L=2, k=2, J_1=2, J_2=4$. Tokens shaded gray are *masked* (i.e., *not* attended to) while those shaded blue are *not masked* (i.e., are attended to).

| Train Context Length (Model) | Eval Length = 64 | Eval Length = 128 |
|---|---|---|
| 64 ($\hat{h}_{\text{short}}$) | 17.59 | 14.74 |
| 128 ($\hat{h}_{\text{long}}$) | 17.62 | 14.49 |

Table 5: Perplexities for (a) model trained on context length of 64 (top); and (b) model trained on context length of 128 (bottom).

length. Accordingly, we also trained a model $\hat{h}_{\text{long}}$ in the same manner as $\hat{h}_{\text{short}}$ but with context length $\bar{L}=128$ (and without PoSE). In Figure 7, we observe that a similar positive correlation is seen (a) for the model $\hat{h}_{\text{long}}$ trained on full contexts of length $\bar{L}$ (Figure 7a), as well as for (b) the model $\hat{h}_{\text{long}}$ trained on contexts of length only $L$ but where the indices $J_{1:k}$ are chosen as in (22) with respect to $\hat{h}_{\text{short}}$ (as opposed to $\hat{h}_{\text{short}}$; Figure 7b), and conversely, the model $\hat{h}_{\text{short}}$ when the indices $J_{1:k}$ are chosen as in (22) with respect to $\hat{h}_{\text{long}}$ (Figure 7c). These observations suggest that the property that a small number $k$ of tokens at positions in $[\bar{L}-L-1]$ can nearly recover the cross-entropy loss at position $\bar{L}$ obtained by training on *all* of the tokens at positions in $[\bar{L}-L-1]$ may be more of a property of the *data distribution*, as opposed to a particularity of any particular language model variant such as $\hat{h}_{\text{short}}, \hat{h}_{\text{long}}$. In particular, it indicates that some property like Definition 3.2 indeed governs the structure of the task of predicting the next token on long contexts.

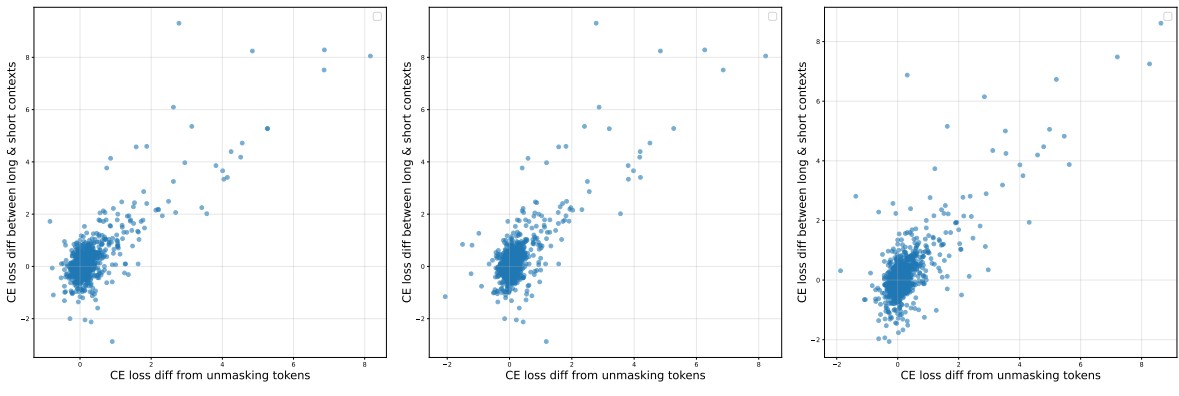

(a) $\hat{h}_{\text{long}}$ CE loss; $J_{1:k}$ chosen from $\hat{h}_{\text{long}}$.   (b) $\hat{h}_{\text{long}}$ CE loss; $J_{1:k}$ chosen from $\hat{h}_{\text{short}}$.   (c) $\hat{h}_{\text{short}}$ CE loss; $J_{1:k}$ chosen from $\hat{h}_{\text{long}}$.

Figure 7: Modification of the plot of Figure 2 where cross-entropy of $\hat{h}_{\text{long}}$ is used instead (Figs. 7a and 7b), and where the opposite model from the one being evaluated is used to generate the indices $J_{1:k}$ (Figs. 7b and 7c).

