# OpenReview forum: "The Role of Sparsity for Length Generalization in LLMs"
_ICML.cc/2025/Conference — ICML 2025 poster_

### Official Review · Reviewer_KgQD · 2025-02-17

**Overall Recommendation:** 3

**Summary:**

This work suggests that length generalization occurs as long as each predicted token depends on a small (fixed) number of previous tokens. This work also conducts experiments on synthetic tasks and natural language.

**Claims And Evidence:**

Yes, the experiment results support the claims and evidence

**Essential References Not Discussed:**

It seems that the key contribution is not new, which is discussed in the following:

[1] Fang, L., Wang, Y., Liu, Z., Zhang, C., Jegelka, S., Gao, J., ... & Wang, Y. (2024). What is Wrong with Perplexity for Long-context Language Modeling?. arXiv preprint arXiv:2410.23771.

[2] Zheng, C., Gao, Y., Shi, H., Xiong, J., Sun, J., Li, J., ... & Li, Y. (2024). DAPE V2: Process Attention Score as Feature Map for Length Extrapolation. arXiv preprint arXiv:2410.04798.

**Experimental Designs Or Analyses:**

I have checked the soundness/validity of experimental designs and analyses

**Methods And Evaluation Criteria:**

Yes, the syntheic task and nature language part results make sense.

**Other Comments Or Suggestions:**

N/A

**Other Strengths And Weaknesses:**

Weakness:
* The key contribution of the work is that length generalization occurs as long as each predicted token depends on a small (fixed) number of previous tokens. **However, it seems that the contribution is not new**, and it has already been discussed in the previous works [1,2]. Therefore, the author should clearly provide the difference between this work and previous work [1,2]
* Also, the core method, presented in Figure 6, is just the sparse attention, which is well-discussed in previous works [3,4].
* Therefore, the author should clearly distinguish their contribution with the previous works.



[1] Press, O., Smith, N. A., & Lewis, M. (2021). Train short, test long: Attention with linear biases enables input length extrapolation. arXiv preprint arXiv:2108.12409.

[2] Fang, L., Wang, Y., Liu, Z., Zhang, C., Jegelka, S., Gao, J., ... & Wang, Y. (2024). What is Wrong with Perplexity for Long-context Language Modeling?. arXiv preprint arXiv:2410.23771.

[3] Ma, X., Liu, Y., Liu, J., & Ma, X. (2025). Mesa-Extrapolation: A Weave Position Encoding Method for Enhanced Extrapolation in LLMs. Advances in Neural Information Processing Systems, 37, 81389-81436.

[4] Lou, C., Jia, Z., Zheng, Z., & Tu, K. (2024). Sparser is faster and less is more: Efficient sparse attention for long-range transformers. arXiv preprint arXiv:2406.16747.

**Questions For Authors:**

The key contribution of the work is not new and has already been discussed in previous works. The authors have to give a response to such limitation.

**Relation To Broader Scientific Literature:**

This work finds that length generalization occurs as long as each predicted token depends on a small (fixed) number of previous tokens.

**Theoretical Claims:**

I have checked the proofs.

---

> ### Author Rebuttal · Authors · 2025-03-28
>
> Thank you for your comments. Below we contrast our work to each of the papers you mentioned. We want to emphasize that with the exception of the 4th paper below (whose theory is different from ours), none of them provide any theoretical analysis, unlike our paper. Moreover, we wish to emphasize that our core method is neither presented in Figure 6 nor is sparse attention. Rather, our main contribution is to explain, using both theory and experiments, why having a sparse dependency structure in data can allow length generalization. We also note that it does not appear that the experiment presented in Figures 2 \& 6 is present in any of the papers you pointed out.
>
> 1. *``What is Wrong with Perplexity for Long-context Language Modeling?''* While the quantity $LSD_\theta(x_i)$ from this paper (which measures the difference in log-likelihoods of the current token using long and short context windows, and is essentially equivalent to what we call $L_{{long}} - {L}_{{short}}$) also plays a role in our paper, a key aspect of our experimental results is that of the tokens in the longer context window, a small number of them can be used to predict the current token $x_i$ nearly as well as all of the long-distance tokens. This ``sparse dependency pattern'' is not highlighted in this prior work.
>
> 2. *``DAPE V2: Process Attention Score as Feature Map for Length Extrapolation.''* While the main method proposed in this paper, DAPE, may be motivated by similar considerations regarding the sparse dependency structure of many tasks (such as associative recall), we emphasize that the content of this paper, which proposes a new empirical method based on applying an MLP to attention patterns so as to improve length extrapolation, is entirely different from ours.
>
> 3. *``Train short, test long: Attention with linear biases enables input length extrapolation.''* We already cite this paper (which introduces ALiBi). We would like to highlight one key difference between our message and that of the ALiBi paper: whereas ALiBi biases the model to put less attention on tokens far in the past, one of our main messages is that length generalization can occur in the (orthogonal) situation when the current token depends on a small number of tokens far in the past.
>
> 4. *``Mesa-Extrapolation: A Weave Position Encoding Method for Enhanced Extrapolation in LLMs.''* This paper has some theoretical analysis showing that certain transformers which fail to length extrapolate for NoPE or ALiBi can length extrapolate for some modified positional encoding schemes introduced therein (namely, Weave PE Extrapolation and Mesa Extrapolation), as well as supporting experiments. These contributions are entirely different from our own.
>
>   5. *``Sparser is faster and less is more: Efficient sparse attention for long-range transformers.''* Using sparse attention, as proposed in this paper, may be motivated by observing that the attention patterns of many actual transformers is sparse, which is closely related to the sparse dependency structure in the data which is the main focus of our paper. Nevertheless, our contribution is *not* to (re-)introduce sparse attention, but rather explain why such a sparse dependency structure suffices for length extrapolation using both theory and experiments.

---

### Official Review · Reviewer_J64p · 2025-03-11

**Overall Recommendation:** 4

**Summary:**

This paper studies why LLMs sometimes can or cannot generalize to input sequences longer than those they were trained on. The authors propose a theoretical framework centered on the concept of "sparsity": the idea that good length generalization occurs when each predicted token depends on only a small, fixed number of previous tokens rather than the entire context.

The paper shows that "locality" (tokens being close together) is also important for length generalization, but this requirement can be mitigated using position coupling techniques, and it introduces "Predictive Position Coupling", an extension of position coupling that allows the model to predict the position IDs for tokens dynamically rather than having them fixed.

**Claims And Evidence:**

The claim that natural language exhibits the sparse planted correlation property is only partially demonstrated. Their experiments show promising evidence, but a more comprehensive analysis of different types of linguistic dependencies and their relationship to length generalization would be needed to fully support this claim.

**Essential References Not Discussed:**

N/A

**Experimental Designs Or Analyses:**

- The paper doesn't fully explain how the "k influential tokens" in the natural language experiments are selected. This is crucial for interpreting the results in Figure 2.
- The experiments don't thoroughly explore how the relationship between sparsity and length generalization might change with model scale, which is important for practical applications.
- While the paper includes confidence intervals for synthetic tasks, a more thorough statistical analysis would strengthen the conclusions, particularly for the natural language experiments.

**Methods And Evaluation Criteria:**

The proposed methods and evaluation criteria seem sound.

**Other Comments Or Suggestions:**

this is just a suggestion, but i personally find a bit uncomfortable to read the paper given that all the references are in a very bright blue. i'd suggest changing the color in a less bright one.

**Other Strengths And Weaknesses:**

other strenghts:

- The introduction of Predictive Position Coupling is an important contribution that extends position coupling to more general settings where the mapping between positions is input-dependent. This broadens the applicability of position coupling techniques.
- The findings have clear practical implications for designing models that generalize better to longer sequences, suggesting that architectures encouraging sparse dependencies might perform better in length generalization tasks

Other weaknesses:
- While the paper includes experiments on natural language data, this section is less developed than the synthetic task experiments. The measurement of sparsity in natural language contexts could be more thorough, with more detailed analysis of how different types of linguistic dependencies affect length generalization.

**Questions For Authors:**

N/A

**Relation To Broader Scientific Literature:**

In the broader landscape of LLM research, this paper helps bridge the gap between the theoretical understanding of transformers and practical techniques for improving their capabilities, particularly regarding handling long contexts efficiently.

**Theoretical Claims:**

- Definition 3.2 (k-sparse planted correlations) - the assumption that non-relevant tokens are i.i.d. from a background distribution is acknowledged as unrealistic. This simplification might affect how well the theory translates to real-world language data.

---

> ### Author Rebuttal · Authors · 2025-03-28
>
> Thank you for your positive comments!
>
> Regarding the how the $k$ influential tokens are chosen: please see the discussion around Eq.~(20) in the appendix.

---

> > ### Comment · Reviewer_J64p · 2025-04-06
> >
> > I thank the authors and after reading their response i decided to keep my overall score to 4.

---

### Official Review · Reviewer_2aUD · 2025-03-12

**Overall Recommendation:** 4

**Summary:**

This work mathematically studies the context length generalization for a single next-token prediction task. It obtains a length generalization error upper bound when the task satisfies certain sparsity (i.e., only a subset of tokens in a context is important to solve the task), locality (i.e., the important tokens must be close to each other), and regularity (i.e., distributions of important token positions do not change a lot with respect to the context length). While locality might be difficult to be satisfied for general next-token prediction tasks, this work also shows that a task-specific position ID re-assignment, Position Coupling, can resolve the issue.

On the empirical side, first, the general theoretical findings (without position coupling technique) are verified on the sparse parity task. To develop the idea of position coupling and mitigate its limitation, a variant called Predictive Position Coupling (PPC) is proposed; the efficacy of PPC is tested for parity (with scratchpad) task and variable assignment task. Lastly, the interplay between sparsity and length generalization capability is studied for the natural language dataset.

**Claims And Evidence:**

The theoretical claims (mainly, Theorem 4.3 and Proposition 4.4) are clearly supported by their proofs provided in Appendices C and D. The authors’ main insight is also tested by several experiments. However a particular claim that “PPC relaxes the downside of position coupling” doesn’t seem to be well supported by any evidence because of lack of PPC’s implementation details: I hope to see a (sub-)section of appendix dedicated to more detailed descriptions about the implementation, training, and evaluation methods using PPC.

**Essential References Not Discussed:**

I can’t find any missing essential references.

**Experimental Designs Or Analyses:**

I checked the experimental designs and their results. The experiments mostly support the main insight of the paper well, while missing some minor details or discussions.

- I can’t find whether PPC doesn’t require the task structure at all. Does the proposed method never require the coupled position IDs in test time? How about in training time? Moreover, doesn’t the training on the prediction of coupled position IDs cause an additional axis of length generalization? If it does, why does the position ID prediction work well even at evaluation time, while the prediction of new next position ID might depend on every previously assigned position IDs (which does not seem to be a ‘sparse’ task)?
- Although not being a *necessary* analysis, “Needle in a Haystack” benchmark might be an extreme case where the authors’ claim should hold. However, they did not discuss the relationship between this task and their claim in the paper. Note that, in fact, several LLMs fail in this benchmark.

**Methods And Evaluation Criteria:**

The paper has chosen several appropriate tasks to examine and verify their claim about crucial factors in achieving length generalization.

**Other Comments Or Suggestions:**

See above.

**Other Strengths And Weaknesses:**

**Strengths**

- I love the paper explaining why position coupling can improve length generalization; it removes the necessity of locality assumption, allowing more tasks to be length-generalizable. It aligns well with the intuition on the benefit of position coupling explained in Cho et al (2024b), adding mathematical rigor significantly.

**Weaknesses**

- The relativity of the class $\mathcal{G}^{\sf key}$ is a bit confusing. In case of $k=1$, the definition of relativity will assign the same attention score to every token; is this realistic? Also, I hope the paper will explain more clearly about why the Item 2 of Assumption 3.2 models the usage of “relative position information”.
- Be consistent with the jargon: there are both “position coupling” and “positional coupling” in the main text. It’d be better to choose only one of them.

**Questions For Authors:**

Authors can find my questions above (sorry for not numbering them…). I am willing to update my evaluation if all the concerns get resolved.

**Relation To Broader Scientific Literature:**

The key contribution of this work is highly related to scaling up the sequence-to-sequence models (like language models based on decoder-only Transformers).

**Theoretical Claims:**

I checked almost every math in all proofs. Most of them seem “correct”, but I mainly am worried about the significance (or tightness) of the given error bound. Below, I list all my concerns about the theoretical parts of the paper.

- The $L$ factor in the bound looks like appearing due to the very first step of the proof: when bounding the risk of $\hat{h}$ for trained lengths $\ell \le L$ by $L\delta$. This seems like an artifact of the analysis, especially caused by $\hat{h}$ defined with an argmin of an expectation. Why don’t we re-define it as a minimax-type estimator, to consider the case where the in-distribution (ID) generalization is already done quite well? For example: $\hat{h} = \arg\min\_{h\in \mathcal{H}} \max\_{\ell\in [L/2, L]} \mathbb{E}\_{({\bf X},{\bf Y})\sim P\_\ell} [\mathcal{L}(h({\bf X}),{\bf Y})].$ In this case, we have a $\delta$-bound instead of $L\delta$-bound:

    $$
    \begin{align*} \mathbb{E}\_{({\bf X},{\bf Y})\sim P\_\ell} [\mathcal{L}(\hat{h}({\bf X}),{\bf Y})] &\le \max\_{\ell'\in [L/2, L]} \mathbb{E}\_{({\bf X},{\bf Y})\sim P\_{\ell'}} [\mathcal{L}(\hat{h}({\bf X}),{\bf Y})] \\\\ &\le \max_{\ell'\in [L/2, L]} \mathbb{E}\_{({\bf X},{\bf Y})\sim P\_{\ell'}}  [\mathcal{L}(h^\star({\bf X}),{\bf Y})] \\\\ &\le \delta.\end{align*}
    $$

    I am a bit concerned about the $L$ factor because the current theorem tries to claim that the error bound may increase as we elongate the trained sequences. It is quite opposite to a common empirical observation that the length generalization capability grows as we train a model on longer sequences. But good news, this issue might be able to be relaxed with a different definition of $\hat{h}$! Still, the parameter $\eta_L$ may scale with $L$ in worst case, so the problem is not completely gone.

- I don’t think the regularity assumption (Ass. 4.2) immediately implies, together with Def 3.2, that the density ratio bounds in math equations roughly in line 885, which is just above Equation (5). I’d be happy if a slightly more detailed derivation is shown in the proof (like in Equation (10)).
- Typo: In a math equation (see line 885; above Equation (5)), $\mathcal{P}_{L-L_{\sf local}}(\mathbf{X}_{1:L-L_{\sf local}}, S^{\star})$  seems correct.
- There seem to be several errors in indices, especially after Equation (8): $t_0 + t_1$ or $t_0+t_1+2$?
- I don’t think the risk bound provides any useful insight about the $L_{\sf local}$-dependency. By merely following the proof, it seems that the error bound is proportional to $(\bar{L} / L_{\sf local})^2$. But is that all? I don’t think so, because it is difficult to understand that the length generalization capability may degrade as the locality gets better (i.e., $L_{\sf local}$ gets smaller). In fact, $\eta_{\bar{L}}$ also depends on $L_{\sf local}$. In my opinion, the theoretical analysis can be stronger by providing an appropriate discussion regarding these.
- One of my biggest concern is the scale of $\eta_{\bar{L}}$. It can be extremely large as it bounds the ratio between two distributions relevant to a very short context length (namely, $L-2L_{\sf local}$) and a large one (namely, $\bar{L}$). If this is the case, can we say that the provided error bound is not vacuous?
- Moving on to the necessity of assumptions (Appendix C.2): I guess the “provable necessity” of the introduced assumptions is an over-claiming since Appendix C.2 does not guarantee that we will fail to discuss length generalizability **every time** we violate one of them. A tone-downed claim might work.
- Minor comment in Def 3.3: I guess $\mathbb{N}^k\times \mathcal{V}^k$ is correct, instead of $(\mathbb{N}\times \mathcal{V})^k$.

---

> ### Author Rebuttal · Authors · 2025-03-28
>
> Thank you for your helpful feedback!
>
> **Regarding the various $L$ factors.** Thank you for these great insights that allow us to improve our bounds!
> Though we did not attempt to optimize these factors in our submission, as you point out, one can indeed improve the dependence on $L$ and $\bar L$, which we discuss in further detail below in response to your questions.
> (For the purpose of interpreting our theoretical results as stated currently, one can think of $\bar L = O(L), \eta_L = O(L), \eta_{\bar L} = O(\bar L)$ (see Remark C.1), and $\delta \leq o(L^{-5})$, which implies a length generalization error (per Theorem 4.3) of $o(1)$.)
>
> - Yes, the factor of $L$ in the error bound can be removed by adopting a minimax objective, as you suggest. (We remark that the objective in Eq.~(3) aligns more closely with what is done in practice, which is to mix over a range of context lengths during training.)
>
> - Yes, the factor of $\bar L^2$ in the error bound can be replaced with $(\bar L / L_{{local}})^2$. The intuition here is that the proof is using $L_{local}$-locality to incorporate the longer context in ``chunks'' of size $L_{local}$. Since there is added error each time we add a chunk, the overall error will scale with $\bar L/L_{local}$. Thus, to minimize the error we want to maximize the size of the chunks. The largest possible value of $L_{local}$ which  still satisfies the assumption of the theorem statement is $L_{local} = L/4$; by choosing this value of $L_{local}$, we can further improve the error dependence to $O((\bar L / L)^2)$. (Essentially, the current counterintuitive bound occurs because the technique used in the proof with values $L_{local} < L/4$ is a suboptimal approach to bound the error, which becomes less suboptimal as $L_{local}$ increases).
>
> - Regarding the size of the parameters $\eta_L, \eta_{\bar L}$, please see Remark C.1, which gives reasonable conditions under which we have $\eta_\ell = O(\ell)$ for all $\ell$. Note  that, for a fixed choice of distributions $Q_\ell^{pos}$, increasing the parameter $L_{local}$ can only make it easier to satisfy Assumption 4.2 for a given choice of $\eta_{\ell}$.   (That said, if we maximize over distributions $Q_\ell^{pos}$ supported on sets $S^\star$ of a given locality $L_{local}$, then the worst-case value of $\eta_\ell$ may increase with increasing values of $L_{local}$.)
>
> - We remark that the dependence on $\eta_L, \eta_{\bar L}$ in the overall error bound of Theorem 4.3 can also be removed under appropriate assumptions: for instance, if we make the distributional assumption of Remark C.1, then the instance of $\eta_L$ in Line 885 can be replaced with $O(1)$ since it is comparing between lengths $L$ and $L - 2L_{local} \geq L/2$. Under the same assumption, the instance of $\eta_{\bar L}$ in Eq.~(10) can also be replaced with $O(1)$ since it is comparing between lengths $\bar L$ and $L - 2L_{local}$; the former length $\bar L$ is larger and on the numerator, meaning that the distribution $Q_{\bar L}^{pos}$ is more ``spread out'' and will put less mass on any $S^\star$ under the assumption in Remark C.1.
>
> - Summarizing, if we incorporate all of the optimizations discussed above, we can decrease our overall error bound from $O(\eta_L \eta_{\bar L} L \bar L^2 \cdot \delta)$ to $O((\bar L / L)^2 \cdot \delta)$, which does not decay with $L$. Hopefully this addresses your concern. We will update the paper to incorporate this discussion.
>
> - We will add more details surrounding the equation around line 885.
>
> **Regarding details of PPC.** The description of PPC can be found on page 7, though we will add a more detailed description. To answer your questions: when discussing PPC, we distinguish between the prompt and the response. During training time, the coupled position IDs for both prompt tokens and response tokens are fed to the model, and the model is trained to predict the coupled position IDs for the response tokens (as well as the actual response tokens). During testing time, the coupled position IDs are only needed for the prompt tokens, but not the response tokens (the model will predict the coupled position IDs for the response tokens).
>
> Yes, training the model to predict position IDs adds an additional axis of length generalization. While the model sometimes makes errors at predicting response position IDs (which we count as an error in our plots), for many of the synthetic tasks that are studied, typically each coupled position ID actually only depends on a small number of previous tokens (e.g., many of the position IDs just increment by 1 at each step). Please see Sections E.2.1 and E.3.1 for the description of our choices of coupled position IDs for our synthetic experiments.
>
> **Regarding Assumption 3.2.** Yes, when $k = 1$, relativity leads to a sort of ``bag of words'' model which is invariant to permutations of the sequence of tokens. We believe the interesting cases occur when $k > 1$.

---

> > ### Comment · Reviewer_2aUD · 2025-04-06
> >
> > Thank you for the author rebuttal.
> >
> > The rebuttal has addressed my concerns about length generalization bound and details about PPC.
> >
> > I raised my score to 4.

---

### Decision · Program_Chairs · 2025-05-01

**Decision:**

Accept (poster)

**Comment:**

The reviewers unanimously endorse this paper, with two reviewers raising their scores after the rebuttal phase. The current scores stand at 4,4,3.

Initial concerns regarding the contextualization of the paper's contributions in relation to existing literature were successfully addressed during the rebuttal. The reviewers provided valuable suggestions for improving the tightness of the theoretical bounds, which the authors have acknowledged and explained in detail how they will incorporate these improvements in the camera-ready version.

I recommend acceptance.